# Membrane curvature sensing of the lipid-anchored K-Ras small GTPase

Hong Liang[1], Huanwen Mu[2], Frantz Jean-Francois[3], Bindu Lakshman[3], Suparna Sarkar-Banerjee[4], Yinyin Zhuang[2], Yongpeng Zeng[2], Weibo Gao[5], Ana Maria Zaske[6], Dwight V Nissley[3], Alemayehu A Gorfe[1], Wenting Zhao[2], Yong Zhou[1]

Plasma membrane (PM) curvature defines cell shape and intracellular organelle morphologies and is a fundamental cell property. Growth/proliferation is more stimulated in flatter cells than the same cells in elongated shapes. PM-anchored K-Ras small GTPase regulates cell growth/proliferation and plays key roles in cancer. The lipid-anchored K-Ras form nanoclusters selectively enriched with specific phospholipids, such as phosphatidylserine (PS), for efficient effector recruitment and activation. K-Ras function may, thus, be sensitive to changing lipid distribution at membranes with different curvatures. Here, we used complementary methods to manipulate membrane curvature of intact/live cells, native PM blebs, and synthetic liposomes. We show that the spatiotemporal organization and signaling of an oncogenic mutant K-Ras$^{G12V}$ favor flatter membranes with low curvature. Our findings are consistent with the more stimulated growth/proliferation in flatter cells. Depletion of endogenous PS abolishes K-Ras$^{G12V}$ PM curvature sensing. In cells and synthetic bilayers, only mixed-chain PS species, but not other PS species tested, mediate K-Ras$^{G12V}$ membrane curvature sensing. Thus, K-Ras nanoclusters act as relay stations to convert mechanical perturbations to mitogenic signaling.

## Introduction

Membrane curvature is a fundamental mechanical property of cells. Various intracellular organelles maintain well-conserved morphologies, defined by distinct membrane curvatures (1–3). On the cell surface, plasma membrane (PM) curvature contributes to vesicular trafficking and determines cell shapes (1–3). Cell morphology changes during mitogen-regulated growth, division, proliferation, and migration (4) and correlates with mitogen-dependent cancer cell transformation and epithelial–mesenchymal transition (5). Over the past 40 years, precise manipulations of cell shapes using micropatterning, microplating, and microneedle methods have consistently shown that the more rounded and flatter mammalian cells undergo more stimulated DNA synthesis, growth, proliferation, and diminished differentiation and apoptosis than the same cells in elongated shapes (6–16). Furthermore, the growth factor–stimulated activation of MAPKs in flatter NIH3T3 cells is rapid and transient but becomes gradual and persistent in the elongated NIH3T3 cells (9). The proliferation rate is sequentially increased in mouse osteoblast cells confined to rectangular, triangular, square, or circular shapes (15). Growth and proliferation of pancreatic, gastrointestinal, breast, and prostate tumor cells display a similar dependence on cell shape (11,17,18). The correlation between mitogen signaling and cell morphology is still poorly understood.

Lipid-anchored Ras small GTPases (including isoforms H-Ras, N-Ras, splice variants K-Ras4A, and K-Ras4B) localize to the cell PM, directly activate MAPK cascade, regulate cell growth/proliferation, and are major drivers of ~1/3 of all human cancer (Fig 1A) (19–21). Ras signaling is mostly compartmentalized to the inner leaflet of the PM, where they anchor mainly using their isoform-specific lipid-modified C-terminal hypervariable regions (22,23). Although Ras proteins lack apparent structural features to detect membrane curvature, previous studies, including ours, have characterized the ability of Ras isoforms to selectively sort distinct lipid head groups and acyl chains in the PM (24–27). Because various lipids display strong curvature preferences (1,2,28,29), the selective lipid-sorting capabilities of Ras isoforms may allow them to sense membrane curvature. Thus, Ras small GTPases are exciting targets for directly mediating mechanotransduction in cancer cells.

To test our hypothesis, we compared the potential membrane curvature–sensing abilities of two main Ras isoforms, H-Ras and K-Ras4B (hereafter simply referred to as K-Ras). We tested the spatiotemporal organization of the full-length oncogenic constitutively active mutants K-Ras$^{G12V}$ and H-Ras$^{G12V}$, as well as that of their respective minimal membrane-anchoring domains, in various cellular and synthetic model systems with different

[1]Department of Integrative Biology and Pharmacology, University of Texas Health Science Center at Houston, Houston, TX, USA   [2]School of Chemical and Biomedical Engineering, Nanyang Technological University, Singapore   [3]National Cancer Institute RAS Initiative, Cancer Research Technology Program, Frederick National Laboratory for Cancer Research, Frederick, MD, USA   [4]Department of Biosciences, Rice University, Houston, TX, USA   [5]School of Physics and Mathematical Science, Nanyang Technological University, Singapore   [6]Internal Medicine, Cardiology Division, University of Texas Health Science Center at Houston, Houston, TX, USA

Correspondence: yong.zhou@uth.tmc.edu

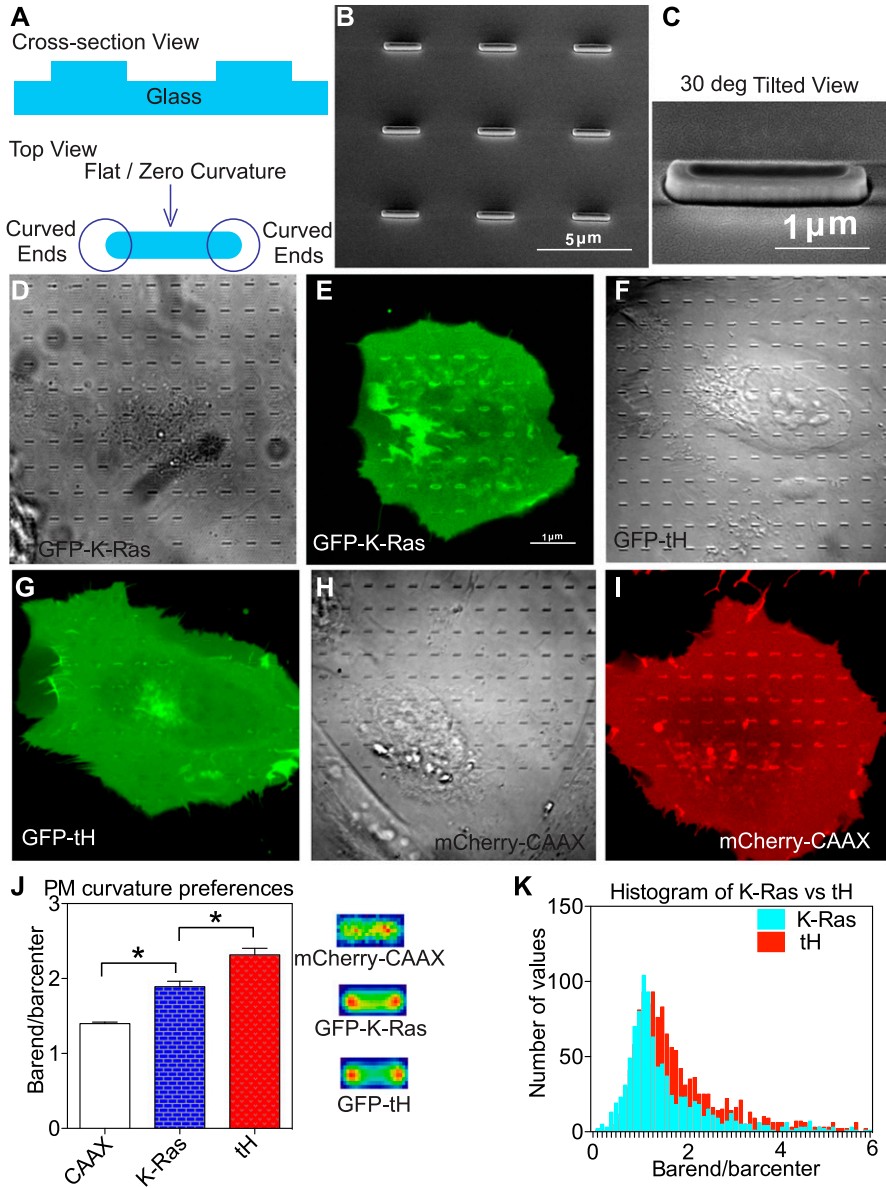

**Figure 1.  Ras localization to the cell PM senses curvature modulations in an isoform-specific manner.**
**(A)** Schematics of side view (top) and top view (bottom) of nanobars etched on the glass surface show that each nanobar possesses two surfaces with distinct curvature: highly curved ends with defined 125-nm curvature radius and flat center region with no curvature. **(B)** A scanning EM (SEM) image shows a SiO$_2$ substrate etched with an array of nanobars with length of 2 $\mu$m and width of 250 nm (125 nm curvature radius). **(C)** A zoom-in SEM image of a single nanobar shows, at a tilted angle, two curved ends (with positive curvature) and a flat center. **(D–I)** U-2OS cells expressing GFP-K-Ras$^{G12V}$ grown over the nanobars are shown in phase contrast (D) and confocal (E), GFP-tH in phase contrast (F) and confocal (G), or mCherry-CAAX in phase contrast (H) and confocal (I). **(J)** Fluorescence intensity ratios between the curved ends and the flat center portions of the nanobars were calculated to indicate the preferential localization of various Ras protein/peptides to the PM curvatures induced by the nanobars. Data are shown as mean ± SEM, with * indicating statistical significance $P < 0.05$ evaluated using the one-way ANOVA. The averaged fluorescence intensities heat maps of all the nanobars imaged are also shown. **(K)** Frequency distribution of all the individual nanobar end/center fluorescence ratios of GFP-K-Ras$^{G12V}$ or GFP-tH is shown. A total of 1,007 nanobars for GFP-K-Ras$^{G12V}$, 1,377 nanobars for GFP-tH, and 415 nanobars for mCherry-CAAX were imaged and calculated.

complexities. Using quantitative imaging methodologies, we show that the flatter membranes with low curvature enhance the PM localization and the nanoclustering of K-Ras$^{G12V}$ and its minimal anchor, while disrupting those of H-Ras$^{G12V}$ and its truncated anchor. Because K-Ras more specifically regulates MAPK during cell growth and proliferation (19–21), our findings are consistent with the long-observed biological phenomenon that MAPK-regulated growth and proliferation are more stimulated in flatter cells than those in elongated cells (6–16). We further explored potential molecular mechanisms and found that the membrane curvature sensing of K-Ras is mediated via the selective sorting of distinct phosphatidylserine (PS) species. Thus, Ras/lipid nanoclusters on the PM act as structural relay stations to convert cell surface morphological modulations into intracellular mitogenic signaling.

# Results

## PM localization of Ras senses PM curvature modulations in an isoform-specific manner

Because Ras signaling is mostly compartmentalized to the cell PM (22,23), we first tested whether the axial PM localization of Ras may be sensitive to PM curvature. We fabricated arrays of vertically aligned nanobars protruding from glass surfaces, similar to our previous studies (Fig 1A–C) (30–32). Each nanobar is 2 $\mu$m long, contains two curved half circles at the ends with a defined 125-nm radius, and a straight line connecting the end circles to provide a flat/zero curvature locally within the same nanobar area (Fig 1A–C). *Homo sapiens* bone osteosarcoma U-2OS cells ectopically expressing a GFP-tagged Ras construct were grown over the

glass surface, with their basolateral PM tightly wrapping around the protruding nanobars and generating positive PM curvature with a precise curvature radius (30–32). The ratio of GFP fluorescence intensity at nanobar-end to center represents the curvature/flat ratio of association. We compared the curvature preferences of three Ras constructs: GFP-K-Ras$^{G12V}$, GFP-tH (minimal anchor of H-Ras), or mCherry-CAAX motif (Fig 1D–I). These constructs were chosen because they sort distinct lipid compositions: K-Ras$^{G12V}$ selectively sorts PS; tH heavily favors cholesterol and phosphoinostol 4,5-bisphosphate (PIP$_2$), whereas the CAAX motif interacts with the membranes randomly and lacks specificity for lipid sorting (24–27,30,33). Averaged fluorescence heat maps and quantification show that mCherry-CAAX, GFP-K-Ras$^{G12V}$, and GFP-tH displayed sequentially higher nanobar end-to-center ratios (Fig 1J). The frequency distribution further shows that the ends/center fluorescence ratios of GFP-tH distributed more toward the curved ends of the nanobars than those of GFP-K-Ras$^{G12V}$ (Fig 1K). Taken together, the axial PM localization of Ras senses membrane curvature changes in an isoform-specific manner: the PM binding of the H-Ras minimal anchor favors more curved PM than that of K-Ras$^{G12V}$.

To further validate the curvature-induced changes in Ras localization to the cell PM, we next performed EM-immunogold labeling (25,26). For these experiments, we used another set of methods to modulate PM curvature: (1) ectopic expression of the membrane curvature sensing/modulating Bin/amphiphysin/Rvs (BAR) domain of amphiphysin 2 (BAR$_{amph2}$) to further induce PM curvature (34,35); (2) acute incubation of cells with hypotonic medium (diluted with 40% deionized water) for 5 min to decrease PM curvature (36). Atomic force microscopy (AFM) was used to obtain the topography of the apical surface of baby hamster kidney (BHK) cells, which is shown to be rough, thus possessing many curved features (Fig S1A–C). To quantify the apical cell surface curvature, we measured the surface roughness, R$_q$, which is the root mean square average of the height deviation from the mean image data plane (37). In Fig S1D, R$_q$ for the unperturbed BHK cell apical surface is ~65 nm. The BAR domain expression further increased R$_q$ to ~102 nm, whereas hypotonic incubation decreased R$_q$ to ~41 nm (Fig S1D). Thus, we effectively manipulated the curvature of cell apical surface: *low* (hypotonic incubation), *medium* (untreated cells), and *high* (BAR domain expression). After acute PM curvature modulations, intact apical PM sheets of BHK cells expressing GFP-tagged Ras were attached to copper EM grids and immunolabeled with 4.5-nm gold nanoparticles conjugated to anti-GFP antibody. The PM sheets were imaged using transmission EM (TEM) at a magnification of 100,000× and the number of gold nanoparticles within a select 1 $\mu m^2$ PM area was counted as an estimate of PM binding of the GFP-tagged Ras (Fig S2A, C, and E). In Fig 2A, with high PM curvature (BAR$_{amph2}$ domain expression), the mean number of gold particles/1 $\mu m^2$ PM area labeling the full-length constitutively active and oncogenic mutant GFP-H-Ras$^{G12V}$ (H-Ras) or that of the truncated minimal membrane-anchoring domain GFP-tH (tH) was higher than those of the oncogenic mutant GFP-K-Ras$^{G12V}$ (K-Ras) or its truncated minimal membrane-anchoring domain GFP-tK (tK). These data are consistent with the nanobar data shown in Fig 1. On the PM with low curvature (hypotonic incubation), the number of immunogold labeling GFP-K-Ras$^{G12V}$ or GFP-tK was significantly

higher than that of the H-Ras constructs, also consistent with the observed Ras isoform-specific curvature preferences in Fig 1. Fig 2C shows the control experiments, where the immunogold labeling was equivalent for all Ras constructs tested in untreated cells with medium extent of PM curvature. In Fig S2J, we replotted the immunogold-labeling data as a function of the apical PM surface roughness (R$_q$): the axial PM localization of GFP-K-Ras$^{G12V}$ or GFP-tK clearly decreased as the PM surface became rougher and more curved, whereas the axial PM binding of GFP-H-Ras$^{G12V}$ or GFP-tH increased as the PM curvature was gradually elevated (Fig S2J). Thus, our nanobar and EM-immunogold-labeling data consistently show that the PM localization of K-Ras favors cell PM with low curvature, whereas that of H-Ras favors more curved PM.

## Ras nanoclustering/oligomerization depends on PM curvature modulations in an isoform-specific manner

Once localized to the cell PM, Ras proteins selectively sort PM lipids to laterally form nanoclusters, in an isoform-specific manner. Specifically, K-Ras nanoclusters are enriched with PS, whereas H-Ras nanoclusters are enriched with PIP$_2$ and/or cholesterol (24–27). The Ras/lipid nanoclusters are functionally important because most Ras effectors contain specific lipid-binding domains and require binding to particular lipids for proper PM localization and activation (38–41). In particular, rapidly accelerated fibrosarcoma (RAF) is a main K-Ras effector and contains separate PS-binding domain (within the cysteine-rich domain, amino acids 1–330) and PA-binding domain (C-terminal amino acids 389–423) (38,42,43). PI3K, on the other hand, is a preferential H-Ras effector and its catalytic subunit p110 binds to PIP$_2$ (44). To quantify the potential sensitivity of Ras nanoclustering to PM curvature modulations, the lateral distribution of the gold particles immunolabeling the GFP-tagged Ras in the same electron micrographs used above in Fig 2A–C was calculated via the Ripley's K-function analysis (25,26,33). Fig S2B, D, and F display the color-coded spatial distribution of the gold nanoparticles in the corresponding EM images in Fig S2A, C, and E, respectively. In Fig S2G, the extent of the nanoclustering, $L(r) − r$, was plotted against the cluster radius, $r$, in nanometers for the corresponding EM images in Fig S2A–F. The $L(r) − r$ value of 1 is the 99% confidence interval (99% CI, the green line), the values above which indicate statistically meaningful clustering. The peak $L(r) − r$ value, termed as $L_{max}$, indicates the optimal nanoclustering (Fig S2H). The extent of nanoclustering is independent of gold labeling densities on the PM for Ras ((45,46) *Preprint*) or other membrane proteins (47). To further validate this independence, we varied the expression levels of GFP-K-Ras$^{G12V}$ on the PM to achieve a wide range of PM-labeling densities. The corresponding $L_{max}$ values were plotted as a function of the gold-labeling densities in untreated BHK cells, which shows no correlation (R$^2$ = 0.0013, pink dashed line indicates the linear regression in Fig S2I). On the PM with high curvature (BAR$_{amph2}$ domain expression), $L_{max}$ of GFP-K-Ras$^{G12V}$ or GFP-tK was markedly lower than $L_{max}$ of GFP-H-Ras$^{G12V}$ or GFP-tH (Fig 2D). On the PM with low curvature (hypotonic incubation), $L_{max}$ of GFP-K-Ras$^{G12V}$ or GFP-tK was significantly higher than that of the H-Ras constructs (Fig 2E). In the control experiments, $L_{max}$ of all Ras constructs was equivalent in untreated cells with medium curvature (Fig 2F). We then re-plotted the $L_{max}$ values

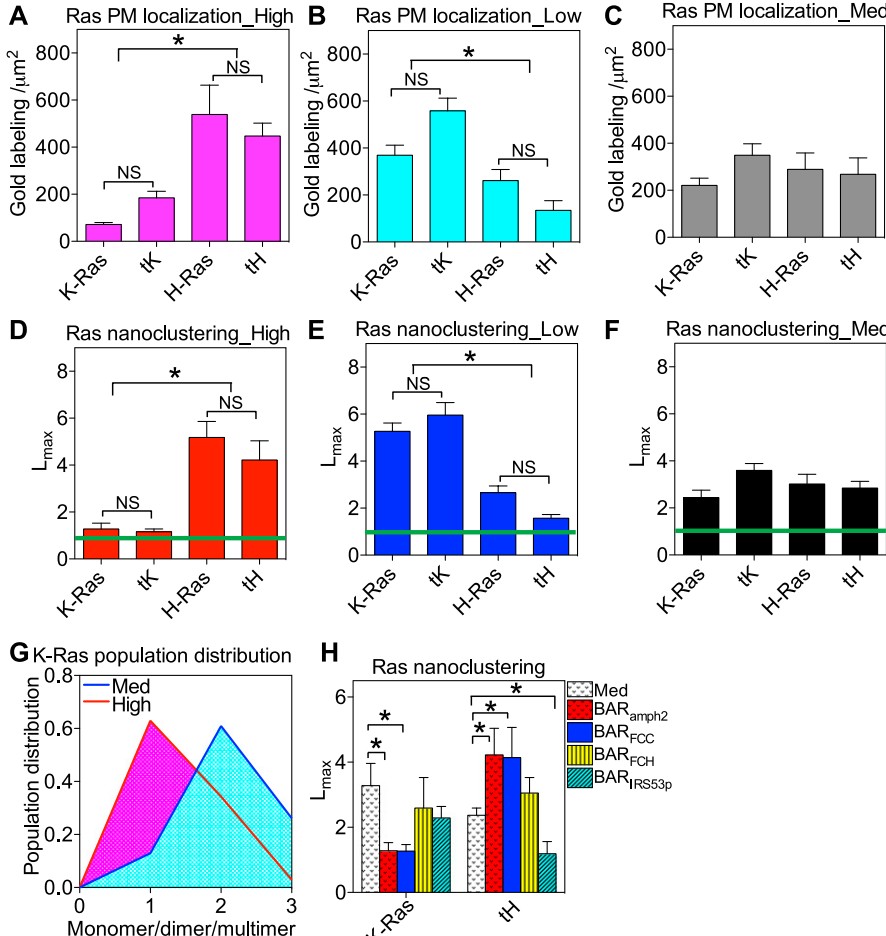

**Figure 2. Ras spatial organization on the PM depends on PM curvature modulations.**
BHK cells were ectopically expressed with GFP-tagged Ras constructs (*K-Ras*: constitutively active oncogenic mutant K-Ras$^{G12V}$; *tK*: the truncated minimal membrane-anchoring domain of K-Ras; *H-Ras*: the oncogenic mutant H-Ras$^{G12V}$; *tH*: the truncated minimal membrane-anchoring domain of H-Ras). Intact apical PM sheets of the BHK cells were attached to EM grids and immunolabeled with 4.5-nm gold nanoparticles conjugated to anti-GFP antibody. **(A–C)** The number of gold particles per 1 $\mu m^2$ area on the intact PM sheets was counted to estimate the localization of the GFP-tagged Ras on the cell PM with high PM curvature (A, ectopic expression of BAR$_{amph2}$ domain), low PM curvature (B, 5-min incubation of cells in hypotonic medium further diluted with 40% deionized water), or with medium curvature (C, untreated cells). **(A–C)** To analyze the lateral spatial distribution of Ras proteins/peptides on the PM, the univariate K-function was used to quantify the nanoclustering of the gold-labeled GFP-Ras constructs in the same EM images used in (A–C). **(D–F)** The optimal nanoclustering, $L_{max}$, for the GFP-Ras constructs in cells with high PM curvature (D), low PM curvature (E), or with medium curvature (F) is shown. The green lines indicate the 99% CI. **(G)** The extent of lateral aggregation of GFP-K-Ras$^{G12V}$ was evaluated using RICS analysis on the PM of live BHK cells co-expressing GFP-K-Ras$^{G12V}$ with either empty vector pC1 (unperturbed PM with medium curvature) or RFP-BAR$_{amph2}$ (high curvature). **(H)** The nanoclustering ($L_{max}$) of GFP-K-Ras$^{G12V}$ or GFP-tH was calculated in BHK cells with unperturbed PM curvature (medium curvature), elevated positive curvature (RFP-BAR$_{amph2}$ and RFP-BAR$_{FCC}$), ineffective truncated BAR (RFP-BAR$_{FCH}$), or elevated negative curvature (RFP-BAR$_{IRS53p}$). All data are shown as mean ± SEM. **(A–C)** In all the gold number counting calculations (A–C), one-way ANOVA was used to evaluate the statistical significance between various groups, with

* indicating $P < 0.05$. **(D–F, H)** In all the nanoclustering analyses (D–F and H), bootstrap tests compared the test samples with 1,000 bootstrap samples and were used to evaluate the statistical significance between different sets, with * indicating $P < 0.05$.

as a function of the apical PM surface roughness ($R_q$): the nanoclustering of GFP-K-Ras$^{G12V}$ or GFP-tK was clearly disrupted as the PM surface became rougher and more curved, whereas the nanoclustering of GFP-H-Ras$^{G12V}$ or GFP-tH was increased as the PM curvature was gradually elevated (Fig S2K). To better examine the extent of the PM curvature–induced changes in Ras population distribution on the PM, we further interrogated our EM-spatial data and calculated the number of gold particles within a distance of 15 nm to generate a population distribution. As shown in Fig S3A–C, elevating the PM curvature significantly increased the number of monomers but decreased the higher ordered multimers of GFP-K-Ras$^{G12V}$. On the other hand, elevating PM curvature decreased monomer population and increased multimer population of GFP-tH (Fig S3D–F). These spatial analyses consistently suggest that Ras proteins sense PM curvature modulations in an isoform-specific manner: K-Ras clustering favors less curved PM, whereas H-Ras clustering prefers more curved PM.

We next tested the potential PM curvature–induced changes in K-Ras clustering/aggregation on the PM of live BHK cells, via Raster image correlation spectroscopy combined with number/balance analysis (RICS-N/B). By counting the fluorescence intensity of GFP-

K-Ras$^{G12V}$, we estimated the population distribution of GFP-K-Ras$^{G12V}$ on the PM of live BHK cells with medium PM curvature (untreated) or with high PM curvature (expressing BAR$_{amph2}$ domain). Higher PM curvature resulted in a marked shift in the population distribution of K-Ras$^{G12V}$ oligomers on the PM (Figs 2G, S3G, and H). Specifically, the presence of the BAR domain shifted the distribution of GFP-K-Ras$^{G12V}$ toward monomers, when compared with the distribution of GFP-K-Ras$^{G12V}$ in untreated cells with medium curvature. The response of GFP-K-Ras$^{G12V}$ to changing PM curvature was consistent on intact PM sheets and live cells.

To further examine whether the responses of Ras nanoclustering to changing PM curvature were dependent upon the specific positive curvature-inducing BAR$_{amph2}$ (an N-BAR) domain used above, we compared and tested another BAR$_{FCC}$ (an F-BAR) domain, which effectively induces a similarly positive PM curvature while sharing little sequence homology with the BAR$_{amph2}$ domain (48). Fig 2H shows that BAR$_{FCC}$ expression disrupted the nanoclustering of GFP-K-Ras$^{G12V}$ and enhanced the clustering of GFP-tH, similar to the effects of BAR$_{amph2}$ domain (Fig 2H). When the BAR$_{FCC}$ is truncated to remove the coiled-coil domain, the resulting BAR$_{FCH}$ domain still binds to the PM but no longer bends the membrane

(48). In parallel EM experiments using BHK cells expressing the BAR$_{FCH}$ domain, the nanoclustering of both Ras isoforms did not respond (Fig 2H). Both BAR$_{amph2}$ and BAR$_{FCC}$ domains induce positive curvature that bends the PM toward the cytosol. Inverse BAR domains, such as BAR$_{IRS53p}$, induce negative curvature and bend the PM outward (49,50). Expression of BAR$_{IRS53p}$ domain did not affect the nanoclustering of GFP-K-Ras$^{G12V}$ but markedly disrupted the clustering of GFP-tH (Fig 2H). Taking together the BAR data, the nanobar data, and the hypotonic flattening data, the response of the Ras nanoclustering to BAR domains is likely PM curvature dependent. Furthermore, Ras spatial distribution may also be sensitive to curvature directions.

### Ras spatiotemporal organization directly responds to changing membrane curvature in live cells, isolated native PM blebs, and synthetic liposomes

Changing cell surface curvature may accompany complex changes in many additional cellular components, such as actin cytoskeleton. To first test whether actin was involved in how K-Ras responded to changing PM curvature in cells, we pretreated BHK cells with either DMSO or 1 $\mu$M Latrunculin A for 5 min to disrupt actin. The two sets of cells were then subjected to PM curvature manipulations (BAR$_{amph2}$ domain expression or hypotonic incubation), followed by EM-spatial analysis. In unperturbed cells with medium PM curvature, Latrunculin A partially abrogated the nanoclustering of GFP-K-Ras$^{G12V}$ (Fig 3A), consistent with previous studies (24,45). Interestingly, the nanoclustering of GFP-K-Ras$^{G12V}$ in cells without or with Latrunculin A treatment responded to changing PM curvature in similar manners (Fig 3A), suggesting that actin does not contribute to the membrane curvature sensing of K-Ras.

To better examine that Ras oligomerization responded directly to changing membrane curvature, we aimed to test the cell PM alone without the presence of intracellular components. For this purpose, we used giant plasma membrane vesicles (GPMVs), which directly bleb from live cells and effectively maintain near-native PM lipid and protein compositions (Fig 3B) (51–53). The GPMV is an ideal system for testing PM properties because they do not contain any cytoplasmic contents, such as cytoskeletal structures and endomembranes (51–53). With only the thin lipid bilayer with low bending stiffness (54,55), these vesicles undergo extensive curvature undulations, which can be modulated by osmotic stresses

(54,56). The GPMVs generated from BHK cells co-expressing GFP-tH with either empty vector pC1 or RFP-tH were subjected to Hepes buffers with different hypertonic (<0 ΔOsm) and hypotonic (>0 ΔOsm) conditions for 5 min. The fluorescence lifetime of GFP was then measured and used to calculate FRET efficiency. Similar fluorescence lifetime imaging microscopy-fluorescence resonance energy transfer (FLIM-FRET) experiments were also conducted using GPMVs containing GFP-tK with either pC1 or RFP-tK. ΔOsm denotes the difference in osmolality between vesicle lumen and the outside buffer. The lipid bilayers under hypertonic stresses undergo predominantly curvature fluctuations, with higher negative ΔOsm values indicating more hypertonic stresses and more extensively folded and curved bilayers (55,56). On the other hand, the lipid bilayers undergo gradual unfolding/flattening at low hypotonic stresses but begin to stretch and generate increasing membrane tension at high positive ΔOsm values (56). Fig 3C shows that the FRET efficiency between GFP-tH and RFP-tH decreased as hypertonic stresses subsided and hypotonic stresses elevated, suggesting that the oligomerization of the H-Ras anchor favors more curved vesicle membranes. Interestingly, the FRET efficiency between GFP-tK and RFP-tK increased as hypertonic stresses subsided and in the low hypotonic regime. Under high hypotonic conditions, the FRET efficiency between GFP-tK and RFP-tK decreased as a function of elevating ΔOsm (Fig 3C). Because the hypertonic regime is dominated by the curvature fluctuations, our data suggest that the oligomerization of the K-Ras anchor favors flatter membranes. The high hypotonic pressure stretched the bilayer, disrupted the global organization of the bilayer, and thus disrupted the oligomerization of both H- and K-Ras anchors.

The GPMVs contain only the PM but still possess high complexity with hundreds of types of lipids and various membrane proteins. To further focus on lipids and test whether K-Ras directly senses bilayer curvature, we used two-component synthetic large unilamellar vesicles (LUVs). With these synthetic vesicles, we had total control of the composition of the vesicular bilayer. The use of polycarbonate membranes with defined pore sizes during the extrusion process also allowed us to generate unilamellar vesicles with defined diameters, which directly and quantitatively correlate with the curvature radius (57,58). Via surface plasmon resonance (SPR), we measured the binding of the purified full-length GTP-bound K-Ras to LUVs composed of 16:0/18:1 PS and 16:0/18:1 PC (POPS/POPC, 20%/80%). As shown in Fig 5A, the membrane

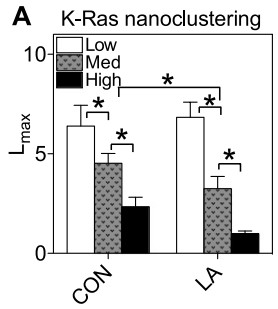
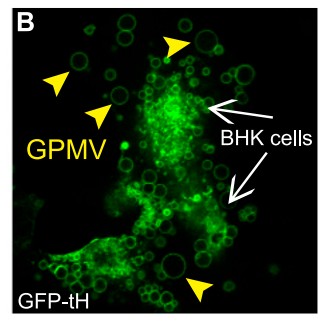
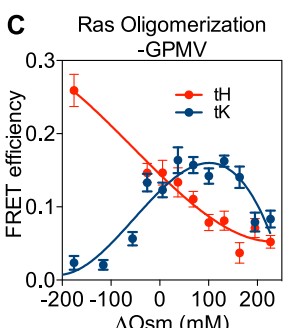

**Figure 3. Ras spatial organization responds to changing membrane curvature in model systems with different complexities.**
**(A)** BHK cells expressing GFP-K-Ras$^{G12V}$ were co-treated with either vehicle DMSO or 1 $\mu$M Latrunculin A for 5 minutes and subjected to PM curvature manipulations as described in Fig 2. Following EM-immunogold- labeling and spatial analysis, the optimal nanoclustering, $L_{max}$, of GFP-K-Ras$^{G12V}$ on the intact PM sheets was plotted as mean ± SEM. **(B)** GPMVs directly blebbed off live BHK cells expressing GFP-tH with empty vector pC1. Each GPMV contains only the plasma membrane with near native lipid/protein composition, but without cytoskeletal structures and endomembranes. **(C)** GPMVs containing GFP-tH (or GFP-tK) with either empty vector pC1 or RFP-tH (or RFP-tK) were incubated in Hepes buffers with different osmolalities to establish various hypertonic and hypotonic conditions across the vesicular bilayers. GFP fluorescence lifetime was measured and used to calculate FRET efficiency as a function of various osmotic stresses, shown as mean ± SEM.

association response unit of the purified K-Ras is termed as $RU_S$, whereas the total liposomal lipid deposition response is termed as $RU_L$. The ratio $RU_S/RU_L$ is plotted as a function of K-Ras concentrations (Fig 5A). Each point depicted in the curves represents the steady-state values of the sensorgram at the corresponding K-Ras concentration, thus showing the binding isotherms. Fig 5A shows that the binding of K-Ras increased as the diameters of the LUVs became larger and the bilayers became less curved. Our K-Ras binding assay strongly suggests that K-Ras membrane binding favors less curved and flatter bilayers. Our data obtained in live cells, intact cell PM sheets, blebbed PM vesicles, and synthetic liposomes are entirely consistent, in that Ras spatiotemporal organization directly responds to changing membrane curvature in an isoform specific manner.

### Spatial segregation of PM lipids senses PM curvature in distinct manners

We next explored a potential mechanism mediating Ras membrane curvature sensing. Selective lipid sorting drives Ras spatiotemporal organization on the PM. Lipid head groups and acyl chains together contribute to their packing geometries, thus their ability to laterally distribute to regions of bilayers with different curvatures and potential membrane curvature preferences (1,2,28,59). However, how the distribution of anionic phospholipids in cells potentially responds to changing PM curvature has not been systematically quantified. To do this, we used EM-spatial analysis (25,26). PM lipid head groups were probed via the GFP-tagged lipid-binding domains in BHK cells (25,26). The PM curvature was modulated as described above: hypotonic medium reduced PM curvature (*Low*), untreated cell contained normal level of PM curvature (*Med*), whereas cells expressing the BAR$_{amph2}$ domain displayed further elevated PM curvature (*High*) (3,36,60–62). The immunogold labeling analysis shows that the nanoclustering (Fig 4A) and gold labeling (Fig 4B) of GFP-LactC2, a specific PS probe, decreased as the PM curvature was gradually elevated. On the other hand, the nanoclustering (Fig 4A) and the PM labeling (Fig 4B) of GFP-PH-PLCδ, a specific probe for PIP$_2$, increased as the PM curvature elevated. GFP-D4H, a specific cholesterol probe, showed a similarly increased clustering (Fig 4A) and PM binding (Fig 4B) upon PM curvature elevation. The spatial distribution (Fig 4A) and PM binding (Fig 4B) of phosphoinositol 3,4,5-trisphosphate (PIP$_3$) did not change upon PM curvature modulations. Although the clustering of PA decreased as the PM curvature was gradually elevated (Fig 4A),

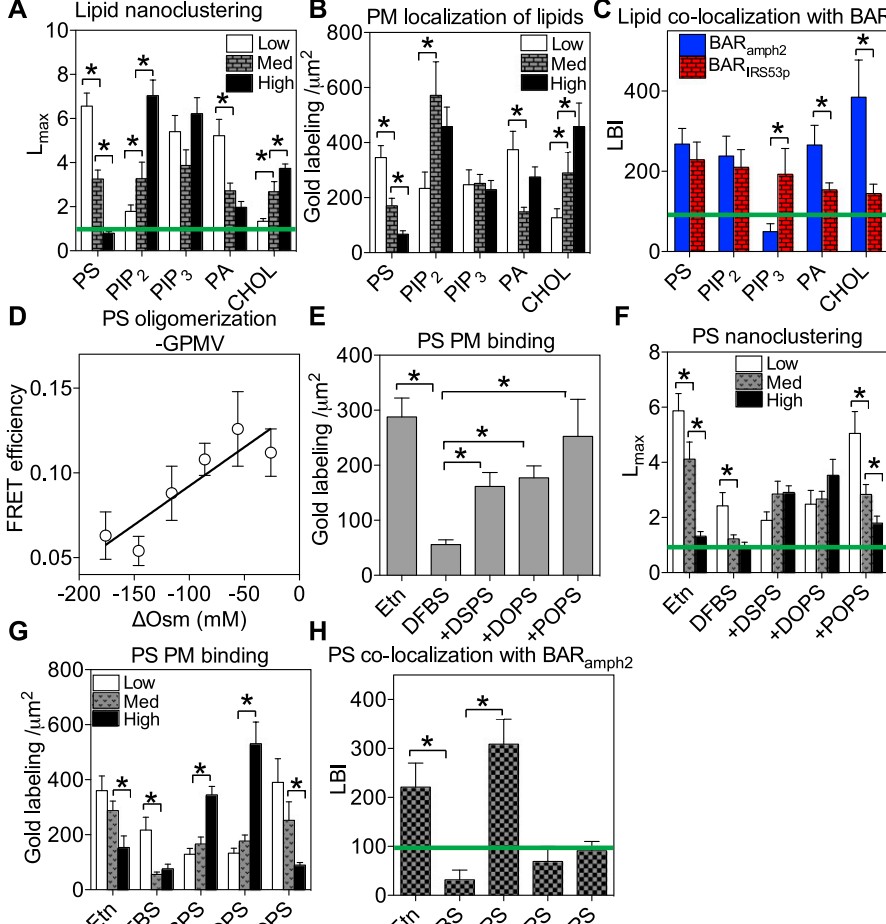

**Figure 4. PM lipids sense membrane curvature modulations in distinct manners.**

**(A, B)** The optimal nanoclustering, $L_{max}$ (A), and number of gold nanoparticles (B) of GFP-tagged lipid-binding domains within the 1 μm$^2$ area on the intact PM sheets of BHK cells with different PM curvature are shown: low PM curvature (hypotonic incubation), medium PM curvature (untreated cells), or high PM curvature (ectopic co-expressing RFP-BAR$_{amph2}$). The green line in (A) marks a $L_{max}$ value of 1 as the 99% CI. **(C)** Co-localization, LBI, between the GFP-tagged lipid-binding domains and the RFP-tagged BAR domains was calculated. The LBI value of 100 indicates the 95% CI (marked by the green line), the values above which indicate statistically meaningful co-clustering. **(D)** The FRET efficiency between GFP- and RFP-tagged LactC2 on GPMVs was calculated using measurements of the fluorescence lifetime of GFP and was plotted as a function of increasing osmolality across the vesicle bilayers (ΔOsm) in the hypertonic regime. **(E)** The immunogold labeling of GFP-LactC2 on the PM is shown when PSA3 cells were under various endogenous PS manipulations: Etn →recovery of the normal endogenous PS; DFBS → endogenous PS depletion; +DSPS → add-back of di18:0 PS under DFBS; +DOPS → add-back of di18:1 PS under DFBS; and +POPS → add-back of 16:0/18:1 PS under DFBS. **(F, G)** $L_{max}$ (F) and PM gold labeling (G) of GFP-LactC2 in PSA3 cells under various PS manipulating and PM curvature modulation conditions was calculated. **(H)** Co-localization, LBI, between the GFP-LactC2 and the RFP-BAR$_{amph2}$ was calculated. All values are shown as mean ± SEM. The statistical significance in the univariate and bivariate spatial analyses (A, C, F, H) was evaluated using the bootstrap tests with * indicating $P < 0.05$. The statistical significance of the gold labeling (B, E, G), as well as the FLIM-FRET data (D), was evaluated using the one-way ANOVA with * indicating $P < 0.05$.

the dependence of PA levels in the PM on the PM curvature modulations was less clear (Fig 4B). $L_{max}$ and gold labeling data for different lipids were also re-plotted as a function of the apical PM surface roughness ($R_q$) (Fig S4A and B), which clearly shows distinct trends in responding to changing PM curvature. Thus, the lateral distribution and PM localization of the anionic PM lipids respond to changing PM curvature in distinct manners.

To further characterize the potential curvature preference of various lipids in the PM, we quantified co-localization between the GFP-tagged lipid-binding domains and the RFP-tagged BAR domains inducing opposite curvature directions: RFP-BAR$_{amph2}$ to induce positive curvature or RFP-BAR$_{IRS53p}$ to induce negative curvature. GFP and RFP on the intact PM sheets attached to the EM grids were immunogold-labeled with 6-nm gold conjugated to anti-GFP antibody and 2-nm gold conjugated to anti-RFP antibody, respectively. The gold nanoparticles were imaged using TEM and the spatial co-localization between the two populations of the gold particles was calculated using the Ripley's bivariate K-function analysis (Fig S4C and D). The extent of the co-clustering, $L_{biv}(r) - r$, was plotted against the cluster radius, $r$, in nanometers (Fig S4E). To summarize the co-localization data, we calculated the area-under-the-curve values for all the $L_{biv}(r) - r$ curves between the cluster radii ($r$) range of 10–110 nm to yield values of L-function-bivariate-integrated (LBI, Fig S4E). The higher LBI values indicate more extensive co-localization, with an LBI value of 100 as the 95% CI (the green line in Fig S4E). PS, PIP$_2$, PA, and cholesterol, but not PIP$_3$, co-localized with BAR$_{amph2}$. All lipids tested co-localized with BAR$_{IRS53p}$. Furthermore, PA and cholesterol co-localized more extensively with BAR$_{amph2}$ than with BAR$_{IRS53p}$ (Fig 4C). We then further validated the ability of the most abundant anionic phospholipid PS to changing membrane curvature, using GPMVs with near-native lipid contents in a FLIM-FRET experiment. Because the hypertonic regime is dominated by curvature fluctuations (56), we subjected the GPMVs to increasing ΔOsm in the hypertonic regime. Fig 4D shows that increasing ΔOsm, which gradually flattened the vesicle membrane, dose dependently elevated the FRET efficiency between GFP- and RFP-tagged LactC2. These data suggest that the lateral aggregation of PS lipids in GPMVs favors flatter and less-curved membranes, consistent with the EM-spatial analysis. Thus, the spatial packing of PM lipids responds to curvature modulations in distinct manners.

To test the curvature preference of lipid tails in intact cells, we also focused on the most abundant anionic lipid PS, using the PS auxotroph PSA3 cells. With a PS synthase (PSS1) knocked down, the PSA3 cells generate significantly less endogenous PS when grown in dialyzed FBS (DFBS) (25,26,63). Supplementation of 10 $\mu$M ethanolamine (Etn) stimulates PSS2 and effectively recovers the wild-type level of the endogenous PS (25,26,63). Immunogold labeling of GFP-LactC2 confirms the changes in PS levels in the PM inner leaflet (Fig 4E). Add-back of different synthetic PS species under the condition of endogenous PS depletion resulted in the selective enrichment of distinct PS species, which has been verified in lipidomics (25) and immunogold labeling (Fig 4E). The nanoclustering (Fig 4F) and PM binding (Fig 4G) of 16:0/18:1PS (POPS) were disrupted upon gradually elevating PM curvature. On the other hand, the clustering of the fully saturated di18:0 PS (DSPS) and the mono-unsaturated di18:1 PS (DOPS) did not respond to changing PM curvature (Fig 4F). Interestingly, the immunogold labeling of GFP-

LactC2 increased markedly upon elevating PM curvature in the presence of either DSPS or DOPS (Fig 4G), suggesting that the PM localization of both PS species favors more curved PM, although their spatial segregation did not change. The $L_{max}$ and gold labeling for various PS species were also plotted as a function of the apical PM surface roughness ($R_q$) (Fig S4F and G). We next tested co-localization of different PS species with the BAR$_{amph2}$-induced PM curvature. Only the fully saturated DSPS co-localized extensively with the BAR$_{amph2}$ domain (Fig 4H). Thus, PS acyl chains are sensitive to changing PM curvature.

### Distinct PS species mediate K-Ras membrane curvature sensing

K-Ras nanoclusters contain distinct PS species, specifically mixed-chain PS species (25). Furthermore, effector recruitment by the constitutively active oncogenic mutant K-Ras$^{G12V}$ occurs only in the presence of the mixed-chain PS, but not other PS species were tested (25). We, here, show that the nanoclustering and PM binding of different PS species responded to changing PM curvature in distinct manners (Fig 4F and G). To test if different PS species selectively mediated K-Ras PM curvature sensing, we compared the binding of the purified full-length GTP-bound K-Ras with different sized LUVs composed of different PS species via SPR (Fig 5A and B). As described above, K-Ras binding increased as the diameters of the LUVs composed of mixed-chain POPS/POPC (20%/80%) became larger (Fig 5A). In parallel SPR experiments, K-Ras binding was independent of vesicle size on LUVs composed of the mono-unsaturated DOPS/DOPC (20%/80%) (Fig 5B). These data clearly suggest that different PS species selectively mediate the membrane curvature–dependent binding of K-Ras. We then tested the PS selectivity on intact cell PM, using the EM-spatial analysis. The endogenous PS depletion in PSA3 cells (DFBS) effectively abolished the PM curvature–dependent changes in the nanoclustering of GFP-K-Ras$^{G12V}$ (Fig 5C). Supplementation of Etn recovered the wild-type PS level and restored the PM curvature sensitivity of GFP-K-Ras$^{G12V}$ (Fig 5C). We then tested the potential selectivity of various PS species in cells via supplementation of synthetic PS species under endogenous PS depletion. Add-back of only POPS, but not DSPS or DOPS, effectively restored the curvature sensitivity of GFP-K-Ras$^{G12V}$ (Fig 5C). Thus, different PS species selectively mediate K-Ras PM curvature sensing consistently in cells and synthetic vesicles.

To further examine whether K-Ras PM curvature sensing is mediated by lipid sorting, we compared the clustering of the unphosphorylated versus phosphorylated K-Ras$^{G12V}$ in BHK cells. Because the K-Ras phosphorylation site Ser181 is located in its minimal membrane anchoring domain immediately adjacent to its polybasic domain (lysines 175–180), phosphorylation of Ser181 alters the conformational sampling of the polybasic domain and switches K-Ras lipid preference from PS to PIP$_2$ (25) without interfering with its enzymatic activities (64,65). As PIP$_2$ local segregation favors more curved membranes, opposite of PS (Fig 4A and B), comparing unphosphorylated versus phosphorylated K-Ras$^{G12V}$ is ideal for examining the potential lipid dependence of K-Ras PM curvature sensing. The nanoclustering and PM binding of the unphosphorylated GFP-K-Ras$^{G12V}$ or an unphosphorylatable mutant GFP-K-Ras$^{G12V/S181A}$ consistently favored less curved PM (Fig 5D and E). On the other hand, the clustering and PM binding of the

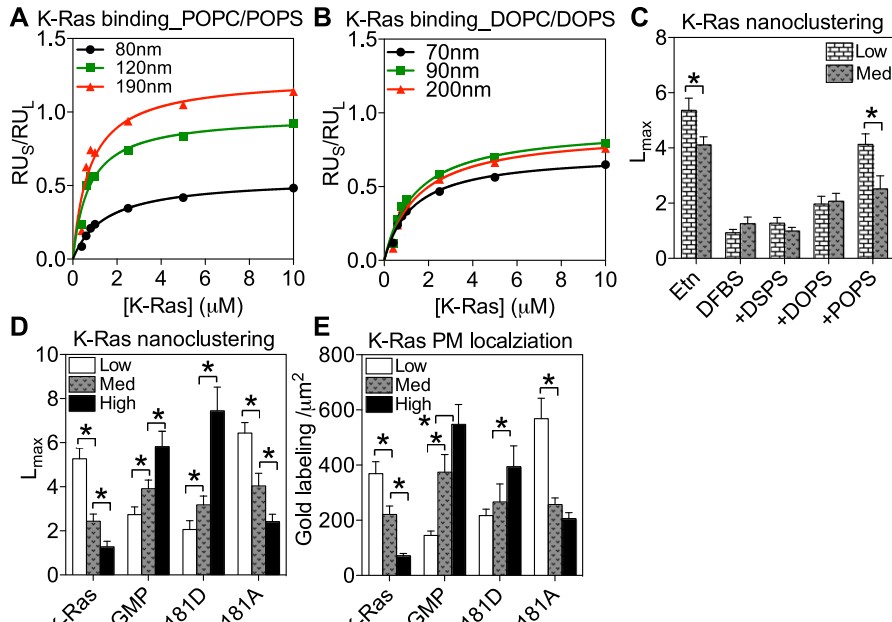

**Figure 5. PS mediates K-Ras PM curvature sensing.**
**(A, B)** Binding of the purified GTP-bound K-Ras to the synthetic two-component liposomes composed of POPC/POPS (80:20) (A) or DOPC/DOPS (80:20) (B) of different diameters was measured in SPR assays. $RU_S$ is the membrane association response unit of the purified K-Ras and $RU_L$ is the total lipid (liposomes of different sizes) deposition response. The ratios of $RU_S/RU_L$ were plotted as a function of K-Ras concentration. The curves shown are binding isotherms, with each point representing the steady-state values of the sensorgram at the corresponding K-Ras concentration. **(C)** PSA3 cells expressing GFP-K-Ras$^{G12V}$ were grown under various PS-manipulation conditions. $L_{max}$ of GFP-K-Ras$^{G12V}$ was analyzed at isotonic (medium PM curvature) versus hypotonic (low PM curvature) conditions. **(D, E)** $L_{max}$ (D) and PM binding (E) of GFP-tagged phosphorylation mutants of K-Ras$^{G12V}$ in BHK cells were quantified upon PM curvature manipulation. All data are shown as mean ± SEM. The statistical significance in the clustering analyses was evaluated using the bootstrap tests, with * indicating $P < 0.05$. The statistical significance of the gold labeling was evaluated using the one-way ANOVA with * indicating $P < 0.05$.

cGMP-induced phosphorylation of GFP-K-Ras$^{G12V}$ (25) or those of a phosphomimetic mutant GFP-K-Ras$^{G12V/S181D}$ were gradually enhanced upon PM curvature elevation (Fig 5D and E). The $L_{max}$ and gold labeling for K-Ras$^{G12V}$ phosphorylation mutants were also plotted as a function of the apical PM surface roughness ($R_q$) (Fig S4H and I). Thus, the phosphorylated K-Ras switches its PM curvature preference to more curved PM. Taken together, K-Ras membrane curvature sensing is largely driven by its ability to selectively sort lipids in the PM.

## K-Ras–dependent MAPK signaling is sensitive to PM curvature modulations

We then tested Ras signaling upon changing PM curvature. Pre-serum-starved wild-type BHK cells were subjected to media with various ΔOsm for 5 min. Increasing ΔOsm across the cell PM gradually flattens the PM. Levels of phosphorylated MEK and

extracellular signal-regulated kinase (ERK) (pMEK/pERK) in the K-Ras–regulated MAPK cascade and the level of pAkt in the H-Ras–regulated PI3K cascade were evaluated using Western blotting. Increasing ΔOsm elevated the levels of pMEK and pERK, but decreased pAkt (Figs 6A and S5A). To test reversibility, BHK cells were incubated in the hypotonic medium for 5 min before switching back to the isotonic medium for various time points. The MAPK and the PI3K signaling recovered to the baseline between 15 and 30 min following switch back to the isotonic medium (Figs 6B and S5B). To further validate the role of K-Ras in the MAPK response to changing PM curvature, we used various Ras-less mouse epithelial fibroblast (MEF) lines, which have all the endogenous Ras isoforms knocked down and stably express a specific Ras mutant (66). We focused specifically on three Ras-less MEF lines: (1) K-Ras$^{G12V}$ MEF line contains only K-Ras$^{G12V}$ but no other endogenous Ras isoforms, (2) BRaf$^{V600E}$ MEF line contains the constitutively active oncogenic mutant of a K-Ras effector BRaf$^{V600E}$ and no endogenous Ras, and

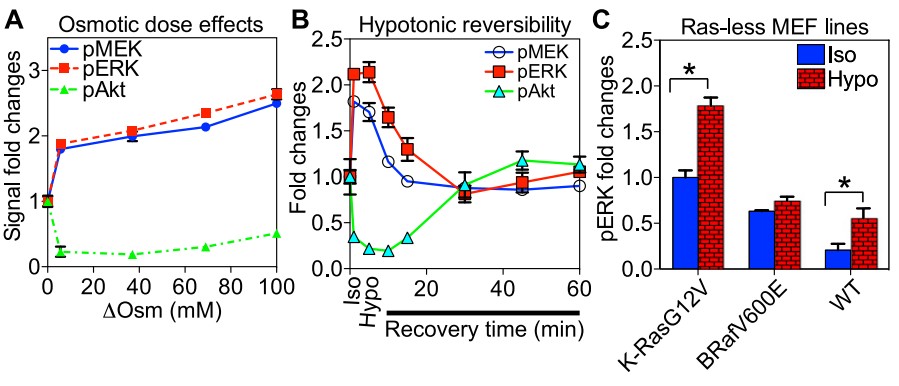

**Figure 6. Ras signaling senses PM curvature modulations in an isoform-specific manner.**
**(A)** BHK cells were pre-serum starved for 2 h and subjected to the media at various osmolality values for 5 min before harvesting. Levels of the phosphorylated MAPK components, pMEK, and pERK were quantified using Western blotting. **(B)** Pre-serum starved BHK cells were incubated in the media prediluted with 40% water for 5 min before being returned to the isotonic medium for various time periods. **(C)** Ras-less MEF lines stably expressing K-Ras$^{G12V}$, BRAF$^{V600E}$, or wild-type MEF were subjected to isotonic or hypotonic (diluted with 40% H$_2$O) Hepes buffers for 5 min before harvesting. For each experiment, the whole cell lysates were used for blotting against pMEK, pERK, and pAkt. Quantitation of three independent experiments is shown as mean ± SEM. All values were normalized against the perspective bands in untreated cells in isotonic medium. The statistical significance was evaluated using the one-way ANOVA, with * indicating statistical significance ($P < 0.05$).

(3) wild-type MEF line contains all the endogenous Ras isoforms (66). The pERK level in the K-Ras$^{G12V}$-MEF increased upon hypotonic condition more efficiently than that in the wild-type MEF line containing endogenous Ras isoforms (Figs 6C and S5C). The MAPK signaling in the BRAF$^{V600E}$-MEF cells no longer responded to the osmotic imbalance. Thus, K-Ras likely drives MAPK response to PM curvature manipulations.

## Discussion

Mitogenic signaling that regulates cell growth, division, proliferation, and migration correlates tightly with the surface curvature-defined cell morphology. Oncogenic mutations in the MAPK cascade play important roles in tumor development, including cancer cell transformation, metastasis, and epithelial-messenchymal transition, which is also tightly correlated with cell morphological changes. This correlation is still poorly understood. Ras proteins are essential upstream regulators of the MAPK cascade and their signaling is mostly compartmentalized to the PM. Our present study, for the first time, compared the potential membrane curvature–sensing abilities of the signaling-essential spatiotemporal organization of Ras isoforms, K-Ras$^{G12V}$ and H-Ras$^{G12V}$, as well as that of their respective truncated minimal membrane anchors tK and tH. We corroborated the membrane curvature–induced responses of Ras proteins in various model systems with a wide range of complexities. Consistent responses of Ras isoforms and their minimal anchors across all systems, from complex live cells to the simplest two-component synthetic bilayers (summarized in Table S1 and Fig S6A) allow us to conclude the following: (1) Ras proteins directly sense membrane curvature in an isoform-specific manner; (2) selective PS species preferentially mediate K-Ras membrane curvature sensing.

K-Ras is the most prevalent Ras isoform in cancer, and its oncogenic mutations contribute to ~80% of all Ras-related cancers (67). The potential ability of K-Ras to sense membrane curvature and to mediate mechanosensing has never been studied until now. Here, we report that the spatiotemporal organization of K-Ras favors flatter membranes with low curvature, consistent in live cells, intact PM sheets, native PM blebs, and two-component synthetic liposomes. The K-Ras–dependent MAPK signaling was also more stimulated in flatter cells (schematic in Fig S6B). These spatial analyses and signaling data suggest that functions of K-Ras, including those of the oncogenic K-Ras mutants, depend on PM curvature modulations during cell shape changes. It has long been observed since the 1970s that the K-Ras–regulated cell growth and proliferation are more stimulated in the flatter cells than the same cells in more elongated shapes (6–16). Furthermore, in intact mouse allographs, melanoma cells expressing a constitutively active K-Ras mutant possess more rounded shape (68), whereas inhibiting the K-Ras signaling in the same melanoma cells alters the cancer cell morphology to a more elongated shape (68). A recent siRNA screening of a large cohort of 92 human pancreatic, lung, and intestinal cancer cell lines also identifies K-Ras mutants as drivers of the epithelial, as opposed to the more elongated, morphology of cancer cells (69). We also observed that the membrane localization and the lateral clustering of the dual-palmitoylated H-Ras prefer

highly curved membranes. The H-Ras–mediated PI3K signaling is also more stimulated when the cell surface was more curved. This is consistent with a previous biological study (11), showing that prostate cells expressing the active H-Ras adopt elongated morphology, whereas the same prostate cells expressing a dominant negative mutant of H-Ras$^{T17N}$ become rounded and flat. Our findings on the PM curvature preferences of tH are also consistent with the previous molecular dynamics simulations, where tH preferentially localizes to the boundaries between liquid-ordered ($L_o$) and liquid-disordered ($L_d$) domains and induce bilayer bending (70). The lateral distribution of the mono-palmitoylated N-Ras anchor tN also depends on bilayer curvature, with more $L_o$ partitioning on the highly curved model bilayers but better $L_d$ affinity on the less curved bilayers (57,58). Thus, our findings provide a potential mechanism for these long-observed mechanosensing events.

In addition to defining cell morphology, Ras membrane curvature preferences may contribute to their distinct intracellular distribution and trafficking. Although all Ras isoforms primarily localize to the PM, they show distinct preferences of intracellular localization (71–73). Specifically, the internalized Ras isoforms quickly dissociate from the endocytic vesicles and become soluble in the cytosol, which is aided via binding to the guanine nucleotide dissociation inhibitor-like solubilizing factor, phosphodiesterase 6δ (PDEδ) (73,74). It has been proposed that the soluble Ras isoforms sample various endomembrane compartments (73,74). H-Ras and N-Ras localize to the Golgi apparatus before being transported to the PM via classic secretory pathways (71–73). On the other hand, K-Ras preferentially localizes to recycling endosomes before catching cargo vesicles back to the PM (71–73). How the different Ras isoforms eventually choose distinct endomembrane compartments is still not clear. Although the distinct hydrophobic and/or electrostatic interactions between Ras isoforms and endomembranes have been attributed (73,74), the potential membrane curvature preferences of Ras isoforms may also contribute to their different trafficking patterns. Various intracellular compartments have distinct morphologies with different curvature magnitudes and directions. Golgi apparatus comprises different structures with highly diverse curvature magnitudes, including highly curved inter-cisternal tubular connections with diameters of ~20 nm and stacks of thin disk-like flat cisternae (75). On the other hand, recycling endosomes contain 80–100-nm diameter vesicular elements and 50–80-nm diameter tubular structures. Endosomal compartments are generally much larger, with diameters ~500 nm (76). The distinct membrane curvature preferences of Ras isoforms may contribute to how they favoriably localize to different intracellular compartments. Indeed, whereas the unphosphorylated K-Ras localizes to the recycling endosomes (73,74), the phosphorylated K-Ras translocates to the Golgi apparatus, similar to H- and N-Ras (65,77). Here, we show that the phosphorylated K-Ras, similar to H-Ras, favors highly curved membranes, whereas the unphosphorylated K-Ras favors membranes with low curvature. Of course, the distinct hydrophobic and/or electrostatic interactions between Ras isoforms and endomembranes may contribute to their membrane curvature sensing, and vice versa.

Although Ras C-terminal membrane-anchoring domains are the main drivers for selective lipid sorting and membrane curvature sensing, polar residues of Ras G-domains also associate with

anionic phospholipids in the membranes. Ras G-domains adopt distinct conformational orientations on the membranes, exposing hydrophobic and charged residues to the bilayers. Specifically, the basic residues in the helix α4 on the GTP-bound H-Ras associate with anionic lipids in the membranes ([78,79]). K-Ras G-domain residues, such as Arg73 and Arg102, associate with PS in the membranes ([80]), whereas several polar residues, such as Lys16, Asp47, and Glu49, of K-Ras G-domain associate with $PIP_2$ (([81]) *Preprint*). AFM experiments also show that the globular K-Ras G-domain is partially embedded in the supported bilayers ([82]). Furthermore, in a signaling complex, K-Ras G-domain and its effector RAF undergo competitive interactions with the anionic lipids in the membranes ([43]). It is clear that the unique conformational orientations of Ras G-domains and their lipid-anchored hypervariable regions, along with the binding of their effectors in signaling complexes, together assume well-defined three-dimensional structures. These conformations are only stabilized when Ras is anchored to the membranes ([43]). Thus, the stabilization of these conformational structures may be optimized by the distinct membrane architectures that match the global shapes of the signaling complexes.

# Materials and Methods

### Materials

GFP-BAR$_{amph2}$, GFP-BAR$_{EFC}$, and GFP-BAR$_{FCH}$ have been generously provided by Dr. Pietro De Camilli at the Yale University.

### Methods

#### EM—spatial analysis
**EM—univariate spatial analysis** The univariate K-function analysis calculates the nanoclustering of a single population of gold immunolabeling on intact PM sheets ([25,26]). Intact PM sheets of cells ectopically expressing GFP-tagged proteins/peptides of interest were attached to copper EM grids. After fixation with 4% paraformaldehyde and 0.1% glutaraldehyde, GFP on the PM sheets was immunolabeled with 4.5-nm gold nanoparticles conjugated to anti-GFP antibody and negative-stained with uranyl acetate. Gold distribution on the PM was imaged using TEM at 100,000× magnification. The coordinates of every gold particle were assigned via ImageJ. Nanoclustering of gold particles within a selected 1 $\mu m^2$ area on intact PM sheets was quantified using Ripley's K-function. The analysis is designed to test a null hypothesis that all points in a selected area are distributed randomly (Equations 1 and 2):

$$K(r) = An^{-2}\sum_{i \neq j}w_{ij}1(\|x_i - x_j\| \leq r), \tag{1}$$

$$L(r) - r = \sqrt{\frac{K(r)}{\pi}} - r, \tag{2}$$

where $K(r)$ indicates the univariate K-function for $n$ gold nanoparticles in an intact PM area of $A$; $r$ is the length scale between 1

and 240 nm with an increment of 1 nm; $\|\cdot\|$ is Euclidean distance; where the indicator function of $1(\cdot) = 1$ if $\|x_i - x_j\| \leq r$ and $1(\cdot) = 0$ if $\|x_i - x_j\| > r$. To achieve an unbiased edge correction, a parameter of $w_{ij}^{-1}$ is used to describe the proportion of the circumference of a circle that has the center at $x_i$ and radius $\|x_i - x_j\|$. $K(r)$ is then linearly transformed into $L(r) - r$, which is normalized against the 99% CI estimated from Monte Carlo simulations. An $L(r) - r$ value of 0 for all values of $r$ indicates a complete random distribution of gold. An $L(r) - r$ value above the 99% CI of 1 at the corresponding value of $r$ indicates statistical clustering at certain length scale. At least 15 PM sheets were imaged, analyzed, and pooled for each condition in the present study. Statistical significance was evaluated via comparing our calculated point patterns against 1,000 bootstrap samples in bootstrap tests ([25,26]).

**EM—bivariate co-localization analysis** Co-localization between two populations of gold immunolabeling GFP-tagged and RFP-tagged proteins/peptides is quantified using the bivariate K-function co-localization analysis ([25,26]). Intact apical PM sheets of cells co-expressing GFP- and RFP-tagged proteins/peptides were attached and fixed to EM grids. The PM sheets were immunolabeled with 2-nm gold conjugated to anti-RFP antibody and 6-nm gold linked to anti-GFP antibody. X/Y coordinates of each gold nanoparticle were assigned in ImageJ, and the co-localization between the two gold populations was calculated using a bivariate K-function. The analysis is designed to test the null hypothesis that the two point populations spatially segregate from each other (Equations ([3]), ([4]), ([5]), and ([6])):

$$K_{biv}(r) = (n_b + n_s)^{-1}[n_b K_{sb}(r) + n_s K_{bs}(r)], \tag{3}$$

$$K_{bs}(r) = \frac{A}{n_b n_s}\sum_{i=1}^{n_b}\sum_{j=1}^{n_s}w_{ij}1(\|x_i - x_j\| \leq r), \tag{4}$$

$$K_{sb}(r) = \frac{A}{n_b n_s}\sum_{i=1}^{n_s}\sum_{j=1}^{n_b}w_{ij}1(\|x_i - x_j\| \leq r), \tag{5}$$

$$L_{biv}(r) - r = \sqrt{\frac{K_{biv}(r)}{\pi}} - r, \tag{6}$$

where $K_{biv}(r)$ is the bivariate estimator comprising two separate bivariate K-functions: $K_{bs}(r)$ describes the distribution of all the big 6-nm gold particles ($b$ = big gold) with respect to each 2-nm-small gold particle ($s$ = small gold); and $K_{sb}(r)$ describes the distribution of all the small gold particles with respect to each big gold particle. The value of $n_b$ is the number of 6-nm big gold particles and the value of $n_s$ is the number of 2-nm small gold particles within a PM area of $A$. Other notations follow the same description as explained in Equations ([1]) and ([2]). $K_{biv}(r)$ is then linearly transformed into $L_{biv}(r) - r$, which was normalized against the 95% CI. An $L_{biv}(r) - r$ value of 0 indicates spatial segregation between the two populations of gold particles, whereas an $L_{biv}(r) - r$ value above the 95% CI of 1 at the corresponding distance of $r$ indicates yields statistically significant co-localization at certain distance yields. Area-under-the-curve for each $L_{biv}(r) - r$ curves was calculated within a

fixed range 10 < *r* < 110 nm and was termed bivariate $L_{biv}(r) - r$ integrated (or LBI) (Equation 7):

$$LBI = \int_{10}^{110} Std \ L_{biv}(r) - r \cdot dr \qquad (7)$$

For each condition, >15 apical PM sheets were imaged, analyzed, and pooled, shown as mean of LBI values ± SEM. Statistical significance between conditions was evaluated via comparing against 1,000 bootstrap samples as described (25,26).

### FLIM-FRET in GPMVs

BHK cells co-expressing GFP-tagged Ras anchors and the empty vector pC1 or RFP-tagged Ras anchors were grown to ~85% confluency and washed with 2× Hepes buffer. The cells were then incubated in Hepes buffer containing 2 mM (N-ethyl maleimide [NEM]) for ~90 min to induce blebbing (83). GPMVs containing GFP- and RFP-tagged Ras anchors (or GFP-tagged Ras anchors with pC1) were then incubated in Hepes buffers containing various percentages of deionized water (hypotonic conditions) or concentrations of NaCl (hypertonic conditions) for 5 min.

GFP fluorescence was visualized using a Nikon TiE wide-field microscope using a 60× oil-emersion PLAN-Apo/1.4 numerical aperture lens (25,26). The fluorescence lifetime of GFP was measured using a Lambert FLIM unit attached to the wide-field microscope. GFP was excited using a sinusoidally stimulated and modulating 3-W 497-nm light-emitting diode at 40 Hz. At least 20 vesicles were imaged and the fluorescence lifetime values were pooled and averaged. Statistical significance was evaluated using one-way ANOVA, with * indicating $P < 0.05$.

### Fabrication and characterization of quartz nanostructures

**Nanofabrication** Nanostructures used in this work were fabricated on the square quartz wafer by using electron-beam lithography. In brief, the chips were spin-coated with 300 nm of positive electron-beam resist PMMA (MicroChem), followed by AR-PC 5090.02 (Allresist). Desired patterns were exposed by EBL (FEI Helios NanoLab), and then developed in IPA:MIBK = 3:1 solution. A 100-nm Cr mask was generated via thermal evaporation (UNIVEX 250 Benchtop) and lift-off in acetone. Nanostructures were subsequently synthesized through reactive ion etching with the mixture of $CF_4$ and $CHF_3$ (Oxford Plasmalab 80). Before cell culture, the nanostructured chips were cleaned by $O_2$ plasma and immersed in Chromium Etchant (Sigma-Aldrich) to remove the remaining Cr mask. SEM (FEI Helios NanoLab) imaging was performed after 10-nm gold coating to measure the dimensional properties of different nanostructures.

**Cell culture** The nanostructure substrates were first coated with fibronectin (2 µg/ml; Sigma-Aldrich) for 30 min at 37°C. After coating, U2OS cell culture was plated onto the substrates with 400,000 cells per 35 mm dish one night before transfection. The cells were maintained in the DMEM supplemented with 10% FBS (Life Technologies) and 1% penicillin–streptomycin (Life Technologies) in a standard incubator at 37°C with 5% $CO_2$ overnight.

**Transfection** For GFP-K-Ras$^{G12V}$, GFP-tH, or mCherry-CAAX transfection in U2OS cells, 1 µg plasmid was mixed with 1.5 µl Lipofectamine 3000 (Life Technologies) and 2 µl P3000 reagent (Life Technologies) in Opti-MEM (Gibco) and incubated for 20 min at room temperature. The medium of the cells to be transfected was then changed to Opti-MEM and the transfection mixture was added. Next, the cells were incubated at 37°C in Opti-MEM. After 4 h, the medium was changed back to regular culture medium and the cells were allowed to recover for additional 4 h before imaging.

**Live-cell imaging and quantification** Live imaging of the transfected U2OS cells on regular nanobar arrays (250 nm wide and 2 µm long) was performed using laser scanning confocal microscopy (Zeiss LSM 800 with Airyscan). In particular, a Plan-Apochromat 100×/1.4 oil objective was used. Excitation of EGFP was performed at 488 nm and detection was at 400–575 nm. Images had a resolution of 1,024 × 1,024 pixels, with a pixel size of 62 nm and a bit depth of 16. During imaging, the cells were maintained at 37°C in an on-stage incubator with FluoroBrite DMEM (Gibco). Z stack images were acquired at 512 × 512 pixels with 100 nm distance between frames.

The curvature preferences of GFP-K-Ras$^{G12V}$ and GFP-tH (Fig 2) were measured on regular nanobar arrays. After averaging, the z-stack images were captured using a confocal microscope; the background intensity of each image was subtracted by a rolling ball algorithm in Fiji with 3-pixel radius. The protein intensities at nanobar ends and centers were quantified using a custom-written MATLAB (MathWorks) code and bar-end/bar-center ratio were calculated and displayed as mean ± SEM as indicated in the figure.

### Raster image correlation spectroscopy—number and balance analysis

Fluorescence measurements for N & B analysis were carried out with Nikon A1 confocal microscope using CFI Plan Apo IR 60× 1.27 NA water immersion objective at 22°C. Live cells were maintained at 37°C in DMEM, 10% bovine calf serum (Hyclone), and 5% (vol/vol) $CO_2$ in a 35-mm glass bottom dish (MatTek Corporation). The cells were imaged in live cell imaging solution containing Hepes buffered physiological saline at pH 7.4 (Life Technologies). Fluorescence images were acquired using a 488-nm laser power at 0.5% that corresponded to 1.75 mW at the measured samples. Rest of the measuring parameters and microscope settings were similar to what has been described in previous report (84). Briefly, for the N & B analysis, acquired images were scanned with 64 × 64 pixel box along the cell margins. Brightness values of the BHK cells expressing mem-EGFP (membrane-bound EGFP, a GFP containing an N terminally palmitoylated GAP-43 sequence cloned in a pEGFP-1 vector) have been used as brightness standards and for calibrating laser power and other microscope parameters. Our data have been analyzed following previous reports (84). Oligomeric size of GFP-K-Ras$^{G12V}$ has been determined as the ratio of the measured brightness and the brightness of monomeric mem-EGFP after subtracting the apparent brightness of immobile molecules.

### SPR

Fully processed KRAS4b was produced and purified as described before (85). All lipids were purchased from Avanti Polar Lipids (Alabaster): 1,2-dioleoyl-sn-glycero-3 phosphocholine (DOPC), 1,2-dioleoyl-sn-glycero-3-phospho-L-serine sodium salt (DOPS), 1-palmitoyl-2-oleoyl-sn-glycero-3-phosphocholine (POPC), and

1-palmitoyl-2-oleoyl-sn-glycero-3-phospho-L-serine sodium salt (POPS).

**Liposome preparation** Lipid stock solutions dissolved in chloroform were mixed at the desired molar composition. Chloroform was evaporated in a gaseous nitrogen stream and afterward dried overnight under vacuum to remove the residual solvent. Before the experiments, the lipids were suspended in ~1 ml 20 mM Hepes buffer (pH 7.4), 150 mM NaCl, 1 mM TCEP, and vortexed to yield a theoretical total lipid concentration of 6 mM. After hydration, the lipid solution was sonicated for 5 min at 25°C and subjected to five freeze–thaw–vortex cycles and another brief sonication. LUVs were formed by extrusion through a polycarbonate filter ranging from 30- to 400-nm pore diameter. The extruded solution was then diluted to a concentration of 1 mM.

**Surface plasmon resonance measurements** SPR experiments were carried out in a Biacore S200 instrument from GE Healthcare. Temperature was set at 25°C for all experiments. A 20 mM Hepes, 150 mM NaCl, pH 7.4, and 5 $\mu$M GppNHp solution was used as running buffer for experiments using K-Ras in the active state. The flow system was primed three times before initiating an experiment. The L1 sensor chip was used in all experiments. The sensor chip surface was rinsed with three injections of 20 mM CHAPS before LUV deposition. 1 mM lipid LUV samples were injected over the L1 sensor chip for 900 s, at a 2 $\mu$l/min flow rate. Loose vesicles were removed with a 36-s injection of 10 mM NaOH at 50 $\mu$l/min. K-Ras at defined concentrations (0.2–50 $\mu$M) were injected over pre-formed lipid vesicle-coated surfaces at 30 $\mu$l/min, for a total of 60 s (association phase). The fully processed K-Ras was soluble in the buffer used for the experiments without detergent. Solutes were allowed to dissociate for 300 s. L1 sensor chip surface regeneration was performed with sequential injections of 20 mM CHAPS (5 $\mu$l/min for 60 s). Baseline response values were compared before and after each experiment to evaluate the effectiveness of the surface regeneration. Raw SPR sensorgram data were collected for both lipid deposition and solute binding. LUV deposition response values were collected from sensorgrams upon reaching a stable response. For each studied molecule, association steady-state response values were collected from individual sensorgrams at $t$ = 60 s. Dissociation response data were collected between 60 and 360 s of each sensorgram.

### AFM
To investigate the topography of the cell apical plasma membrane, BHK cells were grown to a 50% confluency in 60-mm dishes pre-coated with rat tail collagen to enhance cell attachment. After PM curvature manipulations, cells were washed 2× with PBS, fixed in 2.5% glutaraldehyde and maintained in PBS. Random cells were selected, for each treatment, with the aid of optical microscopy (20×). Never-dried cells were scanned in PBS using MLCT cantilevers ($f_0$ = 4–10 kHz, $k$ = 0.01 N/m, ROC = 20 nm) Bruker Corporation. The topography of the cell membrane was determined using contact mode operated in liquid to a scan rate of 0.6 Hz. Images of the cell structure were captured to a scan area of 20 to 50 $\mu$m$^2$, depending on the cell size, using a Bioscope-II atomic force microscope.

AFM was conducted at the University of Texas-Health Science Center AFM Core Facility using a BioScope II Controller (Bruker Corporation). The image acquisition was performed with the Research NanoScope software version 7.30 and analyzed with the NanoScope Analysis software version 1.50 (copyright 2013 Bruker Corporation). This system was integrated to a Nikon TE2000-E inverted optical microscope (Nikon Instruments Inc.) to facilitate bright field and fluorescence imaging of the cell cultures. The roughness of the cell apical plasma membrane was determined by using the roughness command in the Nanoscope Analysis software. The analysis was isolated to a specific area (10 $\mu$m$^2$ box) in the 20 $\mu$m$^2$ scans, to compare only selected portions of an image (same area) in each treatment.

### Western blotting
BHK cells were grown in DMEM containing 10% bovine calf serum to ~85–90% confluency and preserum starved for 2 h before incubation with hypotonic medium containing various percentages of de-ionized water for 5 min. Whole cell lysates were collected. In reversibility experiments, preserum starved BHK cells were exposed to hypotonic medium containing 40% water for 5 min before being returned to isotonic medium for various time lengths before harvesting. Whole cell lysates were used and blotted using antibodies against pMEK, pERK, and pAkt. Actin was used as loading control. Each experiment was performed at least three times separately and data were shown as mean ± SEM. Statistical significance was evaluated using one-way ANOVA, with * indicating $P$ < 0.05.

## Data and Materials Availability

All data are available in the main text or the supplementary materials.

## Supplementary Information

## Acknowledgements

This project was in part supported by the Cancer Research and Prevention Institute of Texas (RP130059, RP170233, and DP150093). This project has been funded in part with Federal funds from the National Cancer Institute, National Institutes of Health, under Contract No. HHSN261200800001E. AFM scanning was conducted at the IM Bioscope II—UT Core Facility as a part of the Internal Medicine Department, University of Texas Health Science Center at Houston.

### Author Contributions

H Liang: data curation and formal analysis.
H Mu: data curation and formal analysis.
F Jean-Francois: data curation and formal analysis.
B Lakshman: data curation and formal analysis.
S Sarkar-Banerjee: data curation and formal analysis.
Y Zhuang: data curation and formal analysis.
Y Zeng: data curation.

W Gao: data curation.

AM Zaske: data curation and formal analysis.

DV Nissley: data curation, formal analysis, and supervision.

AA Gorfe: formal analysis, supervision, and writing—original draft.

W Zhao: data curation, formal analysis, supervision, methodology, and writing—original draft, review, and editing.

Y Zhou: data curation, formal analysis, supervision, funding acquisition, validation, investigation, methodology, project administration, and writing—original draft, review, and editing.

**Conflict of Interest Statement**

The authors declare that they have no conflict of interest.

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
