## [Reviewer comments · Life Science Alliance]

Life Science Alliance

Membrane curvature sensing of the lipid-anchored K-Ras small GTPases

Hong Liang, Huanwen Mu, Frantz Jean-Francois, Bindu Lakshman, Suparna Sarkar-Banerjee, Yinyin Zhuang, Yongpeng Zeng, Weibo Gao, Ana Zaske, Dwight Nissley, Alemaheyu Gorfe, Wenting Zhao, and Yong Zhou

DOI: <https://doi.org/10.26508/lsa.201900343>

Corresponding author(s): Yong Zhou, University of Texas Health Science Center at Houston

Review Timeline:

Submission Date:	2019-02-13
Editorial Decision:	2019-03-11
Revision Received:	2019-06-07
Editorial Decision:	2019-07-01
Revision Received:	2019-07-01
Accepted:	2019-07-02

Scientific Editor: Andrea Leibfried

Transaction Report:

March 11, 2019

Re: Life Science Alliance manuscript #LSA-2019-00343-T

Prof. Yong Zhou
University of Texas Health Science Center at Houston
Integrative Biology and Pharmacology
6431 Fannin Street
MSE R382
Houston, TX 77030

Dear Dr. Zhou,

Thank you for submitting your manuscript entitled "Membrane curvature sensing of the lipid-anchored K-Ras small GTPases" to Life Science Alliance. The manuscript was assessed by expert reviewers, whose comments are appended to this letter.

As you will see, reviewer #1 and #2 think that your conclusions on membrane curvature sensing are not supported by the data provided and that important controls (BAR domain expression inducing curvature) are lacking.

We would thus like to invite you to provide a revised manuscript, addressing these concerns. Importantly, better support for your conclusion that there is a direct link between curvature to Ras recruitment/nanoclustering/signaling strength is needed.

Thank you for this interesting contribution to Life Science Alliance. We are looking forward to receiving your revised manuscript.

Sincerely,

B. MANUSCRIPT ORGANIZATION AND FORMATTING:

Reviewer #1 (Comments to the Authors (Required)):

Yong Zhou and colleagues have long been interested in the interplay between RAS membrane

tethers and the phospholipids that constitute the inner leaflet of the plasma membrane (PM). These interactions promote nanoclustering of the GTPases and thereby modulates signaling. In this study the authors examine the effects of membrane curvature on association and nanoclustering of HRAS and KRAS4B on the PM. They conclude that, whereas KRAS4B prefers less curved membranes, the opposite is true for HRAS and that it is both the phosphatidyl serine (PS) enrichment and, somewhat surprisingly, the acyl chain composition of the PS that regulates this process.

The findings are interesting with regard to membrane biophysics and peripheral membrane proteins. How they might be exploited for targeting RAS signaling remains to be determined. The strength of the paper is the multiple and complementary approaches taken to determine the effects of curvature and the rigorous analysis of each assay.

The biggest weakness is the lack of direct demonstration of membrane curvature. The authors maintain that they could enhance membrane curvature by expressing BAR domains and diminish curvature by applying hypotonic culture conditions. This may well be true. However, it would help if there were some way independent of RAS membrane dynamics to confirm the various states of the cells and relevant PM domains. Since the density and clustering assays rely on a technique whereby the apical membrane is allowed to adhere to copper EM grids and then "ripped off" to expose sheets that can be labeled and analyzed by EM, it is not clear what is meant by curvature since it is the apical surface rather than a curved edge that is examined. Presumably they refer to the overall topology of the apical membrane which is stretched flat under hypotonic conditions but awash with protrusions (e.g. ruffles, lamellipodia, and perhaps microvilli) in the basal state. Precisely how ectopic expression of BAR domains affects the apical membrane is not demonstrated.

Minor issues:

The etched glass experiments shown in Fig. 1 are innovative and interesting. It is not clear whether the etched rectangles are convex or concave with regard to the surface of the cell. Also, it is not clear why more curvature will be induced at the ends of the rectangle. If the PM dives into or is pushed up by the rectangles there should be forced curvature all along, as is indicated by GFP-K-Ras decorating the entire perimeter of each rectangle in Fig. 1C. However, the association with curvature is quantified as fluorescence intensity at barend/barcenter. Some explanation is warranted.

The SPR data shown in Fig. 4 needs some clarification. The Y axis is given as RUs/RUL but nowhere are these values defined. What is shown appears to be a derivative of the raw sensograms. It is also not clear how the analyte, recombinant, fully-processed KRAS4B, was kept in solution without detergent.

Reviewer #2 (Comments to the Authors (Required)):

Using imaging approaches, this group have previously made significant contributions to the understanding of Ras protein and associated lipid nanoclustering at the cell surface. In this manuscript they use their well-established fluorescence and electron microscopy protocols to profile nanoclustering and putative curvature sensing of Ras isoforms in response to alterations to membrane curvature. Whilst the data appear to be internally consistent and suggest that different isoforms show different sensitivities to the different conditions, I am not yet convinced by the interpretations of these data that these reflect different curvature sensing properties that are likely

to be biologically meaningful. I have no doubts that the authors are measuring something interesting, I'm just not sure what it is yet.

Main comments

1. Hypotonic stress to induce flat membranes is also likely to have many other consequences that could influence Ras and lipid distributions and signalling. For example, the actin cytoskeleton that differentially influences Ras isoform nanoclustering is also sensitive to hypotonic conditions. This does not seem like a good model for making very specific interpretations about the role of membrane curvature.
2. The authors are predominantly using an EM based technique for their analysis. This involves ripping the apical surface of cells onto an EM grid. Unless I am mistaken, the nanobars inducing the elevated phenotype are on the basolateral surface. Therefore, I fail to see how the EM analysis of apical membranes can report differences in curvature on a distant area of cell. Similarly, when cells are transfected with BAR domains there is no corroboration that the images being analysed were from cells that expressed the BAR domains. Finally, I also question whether the method is appropriate since the act of ripping the membranes off the cells results in flat sheets of membrane for subsequent analysis. Therefore, any topological differences in curvature are lost during the process. Whilst this is a relatively quick procedure, I don't know how it can be reporting on curvature differences that are no longer evident.
3. The authors conclude that tH/HRAS show higher curvature preference than KRAS based on relative distributions in Figure 1. However, Figure 1E shows that tH is largely excluded from the elevated areas compared to KRAS (Figure 1C). Therefore, whilst there may be a differential association in the tH present, total localisation is very limited and could be interpreted to indicate a relative repulsion from areas of high membrane curvature.
4. Figure 2 is meant to corroborate the findings of Figure 1 however I note the close correlation of nanoclustering data with membrane localisation data. Whilst Ras nanoclustering is consistent over a wide range of expression, at low levels of labelling there is a risk that nanoclustering is not effectively observed. To have confidence in these data I would like to see that the labelling intensity on the cell surface is sufficient to observe differences in nanoclustering ie. do they still see the same differences in nanoclustering between H and K at lower and higher intensities than those in the Figures?
5. I'm not yet convinced about the biological relevance. What % of total area/how often are these curvatures observed on the plasma membrane of cells? In what context? If most Ras still locates to relatively flat areas of cells does this dominate anyway? I can imagine that these areas of curvature are more common on organellar surfaces and so may influence Ras trafficking/endomembrane localisation although this isn't discussed.

Other comments:

1. Differences in membrane curvature also mean differences in membrane tension. Can the authors discriminate between these possibilities?
2. A topological diagram showing the different positive/negative orientations/radii of curvature induced by the different treatments would help the reader.
3. GFP-tagged Ras isoforms differentially locate to filopodia with more obvious KRAS localisation. This seems at odds with the interpretation here that HRAS favours areas of high curvature.
4. The nanoclustering data in Supplementary Figure 1 appears to show that KRAS is nanoclustered over a wide size range suggesting that the labelling is so high that adjacent nanoclusters are also featuring in the analysis. The labelling seems too high if the influence of neighbouring nanoclusters is also featuring so strongly. The elevated condition (Supp Fig 1A/B) also reveals the potential for missing nanoclustering that may in fact be present. The labelling is so low that it is unlikely to report the presence of nanoclusters if they are there.

Reviewer #3 (Comments to the Authors (Required)):

This paper provides a comprehensive report on studies of full length Ras isoforms and C-terminal lipidated peptides on different membrane settings; specifically it is found that K-Ras prefers a level/plane membrane whereas H- and N-Ras prefer a curved membrane. Membrane topology is influenced in several ways. This paper is of general interest for a wide community of protein-membrane scientists and of critical importance and timeliness for the Ras community.

The main points of the paper are strongly supported by the data throughout.

This reviewer has several stylistic issues in that overall the paper is very dense and difficult to read. It should also be considerably shortened and possibly reorganized. A table summarizing all the results presented newly in the paper and those cited from the literature would greatly help.

Some of the language/terms are a little difficult for non-experts to get used to: esp. flatter cells/elongated shapes (where elongated actually means more curvature at the edges). Again, perhaps a table with all the membrane manipulations and outcomes would be useful. Similarly, a overview figures, in the spirit of a table of content schematic would be helpful to visualize the key finding.

Recent literature which shows interactions between the K-Ras G-domain and PIP2 both computationally and experimentally should be referenced somewhere. The comment of a C-Raf membrane PS-binding domain is made but could be more explicit, saying what this domain is. Also, different computational labs. have reported different synergistic/non-synergistic binding effects between the Ras G-domain and the c-Raf CRD.

The discussion section falls a little flat on contemplating the impact of the work. Specifically what are the biological rationales / implications of the different lipid / membrane shape preferences for Ras targeting to membranes and Ras extraction from the membrane or vesicle formation for recycling?

Reviewer #1 (Comments to the Authors (Required)):

Yong Zhou and colleagues have long been interested in the interplay between RAS membrane tethers and the phospholipids that constitute the inner leaflet of the plasma membrane (PM). These interactions promote nanoclustering of the GTPases and thereby modulates signaling. In this study the authors examine the effects of membrane curvature on association and nanoclustering of HRAS and KRAS4B on the PM. They conclude that, whereas KRAS4B prefers less curved membranes, the opposite is true for HRAS and that it is both the phosphatidyl serine (PS) enrichment and, somewhat surprisingly, the acyl chain composition of the PS that regulates this process.

The findings are interesting with regard to membrane biophysics and peripheral membrane proteins. How they might be exploited for targeting RAS signaling remains to be determined. The strength of the paper is the multiple and complementary approaches taken to determine the effects of curvature and the rigorous analysis of each assay.

We thank the reviewer for the kind words.

The biggest weakness is the lack of direct demonstration of membrane curvature. The authors maintain that they could enhance membrane curvature by expressing BAR domains and diminish curvature by applying hypotonic culture conditions. This may well be true. However, it would help if there were some way independent of RAS membrane dynamics to confirm the various states of the cells and relevant PM domains. Since the density and clustering assays rely on a technique whereby the apical membrane is allowed to adhere to copper EM grids and then "ripped off" to expose sheets that can be labeled and analyzed by EM, it is not clear what is meant by curvature since it is the apical surface rather than a curved edge that is examined. Presumably they refer to the overall topology of the apical membrane which is stretched flat under hypotonic conditions but awash with protrusions (e.g. ruffles, lamellipodia, and perhaps microvilli) in the basal state. Precisely how ectopic expression of BAR domains affects the apical membrane is not demonstrated.

To better validate the potential manipulations of plasma membrane (PM) curvature, we used atomic force microscopy (AFM) to image the topography of the apical surface of BHK cells ectopically expressing empty vector pC1, GFP-BAR_{amph2} domain, or pC1 and subjected to acute hypotonic stress (PBS buffer further diluted with 40% ddH₂O) for 5 minutes. After fixation following PM curvature manipulations, we used AFM to scan the apical surface topography (new Fig.S1A-C), and measured the surface roughness, R_q , which is the root mean square average of height deviation taken from the mean image data plane. R_q is an established parameter for characterizing changes in local curvature on the apical cell surface (Dufrene, et al. *Nat Nanotech.* 2017). The apical surfaces of the unperturbed BHK cells were extensively curved, with a mean R_q value of ~65 nm (Fig.S1D). BAR domain expression markedly increased the R_q value to ~109 nm, suggesting a more curved surface. Hypotonic treatment decreased the R_q value to ~33 nm, suggesting a relatively smooth apical surface. Thus, we effectively manipulated cell surface curvature.

We have incorporated the new AFM data in the new Fig.S1 and described the data on Pages 5 and 6 in the main text.

Minor issues:

The etched glass experiments shown in Fig. 1 are innovative and interesting. It is not clear whether the etched rectangles are convex or concave with regard to the surface of the cell. Also, it is not clear why more curvature will be induced at the ends of the rectangle. If the PM dives into

or is pushed up by the rectangles there should be forced curvature all along, as is indicated by GFP-K-Ras decorating the entire perimeter of each rectangle in Fig. 1C. However, the association with curvature is quantified as fluorescence intensity at barend/barcenter. Some explanation is warranted.

We agree that the description of the new nanobar assay was not sufficient earlier. To better explain its working principle, we now include a schematic illustration (Part A in the schematic below), a scanning EM (SEM) image of an array of nanobars (Part B below), as well as an SEM image of a single nanobar in tilted view (Part C below, described on Page 4 and Fig.1A-C in the main text).

The nanobar we fabricated protrudes from the bottom glass surface to push into the cell to generate invaginations on plasma membrane following closely to the geometry of the nanobars, which has been validated using SEM in our previous studies (Zhao, et al. *Nat. Nanotechnol*, 2017; Santoro, et al. *ACS Nano*, 2017; Li, et al. *Nat Protoc*, 2019). With the high-resolution fabrication, we generate two different curvature within one nanobar: two curved ends with defined curvature radius and center flat/straight sidewalls as local reference for zero curvature control. It is worth noting that the x-y plane of nanobar is designed with a half circle at each end with defined radius. Thus, the portion of the PM wrapping around the ends of the nanobars adopts a positive curvature when presented to the intracellular constituents. Therefore, the association with curvature can be quantified as fluorescence intensity at curved ends vs. flat center, i.e. barend/barcenter, which has been demonstrated in our earlier works (Zhao, et al. *Nat. Nanotechnol*, 2017; Santoro, et al. *ACS Nano*, 2017; Li, et al. *Nat Protoc*, 2019).

To better illustrate the working principle of this nanobar assay, we have added text description rewritten as “We fabricated arrays of vertically aligned nanobars protruding from glass surfaces, similar to our previous studies (Fig.1A-C) (Zhao, et al, *Nat. Nanotechnol*, 2017; Santoro, *ACS Nano*, 2017; Li, et al. *Nat Protoc*, 2019). Each nanobar is 2 μm long, contains two curved half circles at the ends with a defined 125 nm radius, and a straight line connecting the end circles to provide a flat / zero curvature locally within the same nanobar area (Fig.1A-C) The ratio of GFP fluorescence intensity at nanobar-end to center represents the curvature / flat ratio of association.” starting from Line 23 on Page 4.

The SPR data shown in Fig. 4 needs some clarification. The Y axis is given as RUs/RU_L but nowhere are these values defined. What is shown appears to be a derivative of the raw sensograms. It is also not clear how the analyte, recombinant, fully-processed KRAS4B, was kept in solution without detergent.

We apologize for this oversight. We have clarified the SPR data in our revised manuscript (Page 11, legends of Fig.5).

Specifically, RU_s is the membrane association response unit of the purified K-Ras and RU_L is the total lipid (liposomes of different sizes) deposition response. For the curves shown in

the plots, the ratios of RU_S/RU_L were plotted as a function of K-Ras concentration. Each point depicted in the curves represent the steady-state values of the sensorgram at the corresponding K-Ras concentration. Thus, the curves shown are binding isotherms.

The fully processed K-Ras is soluble in the buffer used for the experiments without detergent. We have modified the main text to clarify this point in the Method section (Page 26).

Reviewer #2 (Comments to the Authors (Required)):

Using imaging approaches, this group have previously made significant contributions to the understanding of Ras protein and associated lipid nanoclustering at the cell surface. In this manuscript they use their well-established fluorescence and electron microscopy protocols to profile nanoclustering and putative curvature sensing of Ras isoforms in response to alterations to membrane curvature. Whilst the data appear to be internally consistent and suggest that different isoforms show different sensitivities to the different conditions, I am not yet convinced by the interpretations of these data that these reflect different curvature sensing properties that are likely to be biologically meaningful. I have no doubts that the authors are measuring something interesting, I'm just not sure what it is yet.

We appreciate the reviewer's nice comments and understand the reviewer's concern. We are confident of our proposal that Ras proteins directly sense membrane curvature because we observed consistent responses of different Ras isoforms, including their minimal membrane anchors, to changing membrane curvature in multiple model systems with varying degrees of complexities.

We agree with the reviewer that a network of highly complex constituents works together to define the mechanical properties of cells, including the plasma membrane (PM) curvature. Any mechanical perturbation leads to a concerted effort by multiple cellular constituents to respond. Thus, it is nearly impossible to specifically modulate PM curvature in intact cells. We, hence, designed our experiments to correlate potential curvature-dependent Ras behaviors in cells with their responses in the isolated native PM and simplified synthetic liposomes, where we had complete control of their composition and the magnitude of bilayer curvature.

Specifically, in cells, we manipulated PM curvature via nanobars, expression of BAR domains and hypotonic stresses. Despite the complex effects of these manipulations, they share a common feature: changing surface curvature. We have performed extensive tests in the previous and current studies to verify the induced PM curvature changes. The nanobar surfaces with distinct curvatures have been visualized in scanning EM (SEM) (Zhao, et al. *Nat. Nanotechnol*, 2017; Santoro, et al. *ACS Nano*, 2017; Li, et al. *Nat Protoc*, 2019). Here, we have performed additional atomic force microscopy (AFM) experiments to scan the surface topography of BHK cells subjected to PM curvature manipulations (Fig.S1, description on Pages 5 / 6). We then measured the surface roughness, R_q , which is the root mean square average of height deviation taken from the mean image data plane. R_q is an established parameter for characterizing changes in the curvature on the apical cell surface (Dufrene, et al. *Nat Nanotech*. 2017). We, here, show that the apical surfaces of the unperturbed BHK cells had a mean R_q value of ~65nm (Fig.S1D). BAR domain expression markedly increased the R_q value to ~109nm, suggesting a rougher apical surface with more curved features. Hypotonic treatment decreased the R_q value to ~33nm, suggesting a relatively smooth apical surface with less curved features. Thus, our methods effectively manipulated cell surface curvature. Consistent responses by different Ras oncogenic mutants to low, medium or high PM curvature implied a potential membrane curvature sensing ability of Ras isoforms in cells (Fig.1, Fig.2, Fig.S2 and Fig.S3).

To further focus on the PM, we compared the behaviors of Ras truncated minimal membrane-anchoring domains with their respective full-length constitutively active mutants (Fig.2). In the absence of the enzymatic G-domains, these minimal anchors have been considered mainly as membrane-interacting motifs, thus acting as sensors of changing membrane properties. The minimal membrane anchors responded to changing PM curvature in a similar fashion as their respective full-length cognates (Fig.2A-F), further supporting our view that Ras proteins sense membrane curvature. This view was further strengthened by our new EM experiments (Fig.3A), where we observed similar responses of K-Ras nanoclustering to changing PM curvature in cells with intact actin vs. cells with disrupted actin. Taken together, our data in cells suggest that Ras spatiotemporal organization senses PM curvature.

To more exclusively test the PM, we needed to strip away all cytoplasmic constituents and retain only the PM. We, thus, used the giant plasma membrane vesicles (GPMVs) because the GPMVs were blebs generated from the PM of live cells and contained no intracellular organelles and cytoskeletal structures (no actin and no microtubules) (Levental et al. *PNAS*, 2010; Levental, et al. *PNAS* 2011; Sezgin, et al. *Nat Protoc.* 2012; Lorent, et al. *Nat Commun.* 2017). Containing only PM constituents, the GPMVs allowed us to directly examine Ras sensitivities to the PM properties. Thermodynamic calculations and mechanical experiments have established that the thin lipid bilayers, such as those of the GPMVs, undergo extensive folding and unfolding fluctuations (Rawicz, et al. *Biophys J*, 2000; Needham and Nunn, *Biophys J.* 1990; Zhou and Raphael, *Biophys J*, 2005, Zhou and Raphael, *Biophys J*, 2007). Osmotic stresses have been established to effectively modulate the bending fluctuations of membranes (Rawicz, et al. *Biophys J*, 2000; Needham and Nunn, *Biophys J.* 1990; and Ho, et al. *Langmuir*, 2016). Thus, our use of different ΔOsm effectively manipulated the bilayer curvature fluctuations of the GPMV bilayers (Fig.3B and C).

Our cellular and GPMV data were further corroborated by the synthetic liposome data, where the binding of the purified K-Ras depended on the sizes of the 2-component unilamellar liposomes (80% PC and 20% PS) in our SPR assay (Fig.5A and B). These synthetic liposomes were the simplest systems that allowed us to focus squarely on specific lipids and exclusively test bilayer properties. Because of the simple composition and the spherical geometry, the diameters of these synthetic liposomes directly correlate with their curvature, which has been one of the most established methods for manipulating bilayer curvature (Helfrich, *J Phys France* 1986; Drin, et al. *Nat Struc Mol Biol* 2007; Larsen, et al. *Nat Chem Biol* 2015). We synthesize these model liposomes in the lab, thus giving us total control on the lipid compositions and vesicle sizes (fine control of bilayer curvature magnitudes).

Because of the consistent responses of Ras spatiotemporal organization across all these systems, we are confident that our data suggest a direct membrane curvature sensing capability of Ras.

Main comments

1. Hypotonic stress to induce flat membranes is also likely to have many other consequences that could influence Ras and lipid distributions and signalling. For example, the actin cytoskeleton that differentially influences Ras isoform nanoclustering is also sensitive to hypotonic conditions. This does not seem like a good model for making very specific interpretations about the role of membrane curvature.

To further evaluate the potential contribution of actin cytoskeleton in K-Ras curvature sensing in cells, we disrupted actin organization via co-treatment of Latrunculin A. Specifically, we modulated PM curvature of BHK cells expressing GFP-K-Ras^{G12V} without / with the co-treatment of Latrunculin A. Our EM-nanoclustering analysis shows that, in isotonic buffer,

treatment of Latrunculin A partially decreased the clustering of GFP-K-Ras^{G12V} on the PM, consistent with previous findings (Plowman, et al. *PNAS*, 2005; Zhou, et al. *Mol Cell Biol* 2014). Interestingly, gradual elevation of the PM curvature disrupted the nanoclustering of GFP-K-Ras^{G12V} in BHK cells either with or without Latrunculin A. This new data (Fig.3A, Pages 9 and 10) strongly suggests that the PM curvature-induced changes in K-Ras clustering is independent of actin.

Our new Latrunculin A data (Fig.3A) is also consistent with our existing data obtained from other model systems that do not contain actin. For instance, GPMVs have been consistently shown to contain no actin/microtubule structures and no intracellular organelles (Levental, et al. *PNAS*, 2010; Levental, et al. *PNAS*, 2011). The FRET-based oligomerization of K-Ras anchor, tK, favored flattening of membranes in the hypertonic and low hypotonic regimes, while oligomerization of H-Ras anchor, tH, favored more curved membranes (Fig.3B and C). This data is consistent with our findings in live/intact cells. Additionally, our SPR assay shows that the binding of the purified K-Ras favored the larger and flatter two-component synthetic liposomes, also consistent with cell and GPMV data (Fig.5A). Taking all these systems into account, our data consistently suggest that membrane curvature sensing of Ras spatiotemporal organization is independent of actin cytoskeleton.

2. The authors are predominantly using an EM based technique for their analysis. This involves ripping the apical surface of cells onto an EM grid. Unless I am mistaken, the nanobars inducing the elevated phenotype are on the basolateral surface. Therefore, I fail to see how the EM analysis of apical membranes can report differences in curvature on a distant area of cell.

While we used the etched nanobars to manipulate basolateral PM curvature in Fig.1, our EM experiments (Fig.2-5) did not involve any nanobars. In all of our EM experiments, our cells were grown on ordinary glass coverslips and we manipulated the apical PM curvature via ectopic expression of BAR domains, or hypotonic stress. As described above, we have used AFM to carefully validate the ability of BAR domain expression and hypotonic stresses to manipulate the curvature of the apical cell surface (Fig.S1 and description on Pages 5/6). We have clarified this point at the beginning of the second paragraph on Page 5.

Similarly, when cells are transfected with BAR domains there is no corroboration that the images being analysed were from cells that expressed the BAR domains.

To validate that the observed changes in Ras spatiotemporal organization were caused by the expressed BAR domains, we have included various control experiments. Specifically, we compared the Ras spatiotemporal organization in the presence of effective BAR domains, BAR_{amph2} and BAR_{FCC}, with a truncated ineffective BAR domain (BAR_{FCH}) (Fig.2H). We saw consistent responses of Ras constructs to either BAR_{amph2} or BAR_{FCC} domains. On the other hand, none of the Ras constructs responded to the ineffective BAR_{FCH}. In parallel experiments, we also tested an inverse BAR domain (BAR_{IRS53p}, negative curvature), which induces an opposite curvature than the other BAR domains tested. Ras spatiotemporal organization responded differently to BAR_{IRS53p} than other BAR domains, suggesting that we were observing the BAR-induced effects.

The BAR domains induce further cell surface curvature, whereas hypotonic incubation flattens cell PM (verified in our new AFM experiments). We observed opposite responses to BAR expression vs. hypotonic flattening from all the Ras constructs tested. This data further supports that we were observing BAR-induced effects.

In all our BAR domain experiments, we co-transfected BHK/PSA3 cells with a GFP-tagged Ras and an RFP-tagged BAR domain, which has been established to be highly efficient to achieve co-expression of both constructs. Taken together, we are confident that the observed changes in Ras spatiotemporal organization in our experiments were caused by the BAR expression.

Finally, I also question whether the method is appropriate since the act of ripping the membranes off the cells results in flat sheets of membrane for subsequent analysis. Therefore, any topological differences in curvature are lost during the process. Whilst this is a relatively quick procedure, I don't know how it can be reporting on curvature differences that are no longer evident.

We fixed the isolated PM sheets immediately after the rip-off, thus effectively maintaining the spatial localization of various PM components in their native positions. Therefore, we are faithfully reporting the spatial distribution of PM components in their native conditions. Indeed, we have validated our findings in EM rip-off experiments in intact/live cells. Specifically, we used Raster image correlation spectroscopy (RICS) analysis to measure the spatial distribution of GFP-K-Ras^{G12V} in *live BHK cells* without / with RFP-BAR_{amph2} domain. As shown in Fig.2G and Fig.S3G and H, the curvature-dependent changes in the monomer/dimer/multimer populations of GFP-K-Ras^{G12V} in live cells were very consistent with the changes in the monomer/dimer/multimer populations of GFP-K-Ras^{G12V} measured in the EM rip-off experiments (Fig.S3A-C). Thus, we are confident that our rip-off protocol faithfully reports the spatial distribution of PM components in their native conditions.

Our previous studies also strongly support our view. In particular, spatial distribution of many lipids, such as cholesterol, PIP₂ and PA, has been well-established to favor highly curved membranes. In previous studies, we have extensively compared the spatial distribution of these lipids in EM rip-off vs. other quantitative imaging methods in intact/live cells. Specifically, we regularly used EM rip-off and FLIM-FRET (in fixed intact cells) in parallel experiments (Zhou, et al. *Mol Cell Biol*, 2014, Zhou, et al. *Science*, 2015, Zhou et al. *Cell*, 2017). We have also compared the clustering behavior of PIP₂ in the EM rip-off vs. immobile fraction of PIP₂ in the PM of live cells in FRAP experiments (Zhou, et al. *Science*, 2015). In all cases, the spatial distribution of lipids obtained in the EM rip-off method is entirely consistent with their oligomerization/diffusion behavior in intact/live cells. Thus, these data strongly suggest that our EM rip-off faithfully reports the distribution of the PM components in intact/live cells.

Our findings are also supported by the literature, which used the same rip-off protocol to prepare PM lawns to evaluate caveolae, the bulb-like curved structures on the plasma membrane (Fairn, et al. *J Cell Biol*, 2011; Prior, et al. *Nat Cell Biol*. 2001). These studies have consistently identified that the caveolae structures stay intact when using the same rip-off protocol. These findings in the literature support our view that our rip-off protocol effectively maintains the curved membranes.

3. The authors conclude that tH/HRAS show higher curvature preference than KRAS based on relative distributions in Figure 1. However, Figure 1E shows that tH is largely excluded from the elevated areas compared to KRAS (Figure 1C). Therefore, whilst there may be a differential association in the tH present, total localisation is very limited and could be interpreted to indicate a relative repulsion from areas of high membrane curvature.

We feel that comparing total localization on the nanobars (via measuring overall fluorescence intensity on the entire elevated nanobars) may not be appropriate in evaluating potential curvature preferences. This is because overall fluorescence intensity can be influenced by many factors, such as expression level, laser gain and/or exposure time, etc. As such, we have been motivated

to use a fluorescence ratio of the curved ends / flat center on each individual nanobar. This fluorescence ratio compares fluorescence intensities on separate parts of a single nanobar within a single cell, thus eliminating potential artifacts from expression levels, laser powers and exposure time.

4. Figure 2 is meant to corroborate the findings of Figure 1 however I note the close correlation of nanoclustering data with membrane localisation data. Whilst Ras nanoclustering is consistent over a wide range of expression, at low levels of labelling there is a risk that nanoclustering is not effectively observed. To have confidence in these data I would like to see that the labelling intensity on the cell surface is sufficient to observe differences in nanoclustering ie. do they still see the same differences in nanoclustering between H and K at lower and higher intensities than those in the Figures?

To further validate the potential influence of labeling intensities on Ras clustering, we performed additional EM experiments in the untreated BHK cells expressing different levels of GFP-K-Ras^{G12V} to achieve a wide range of gold labeling (Fig.S2I). The L_{max} values for individual EM images were then plotted as a function of gold labeling intensities. We only compared untreated BHK cells because additional perturbations may simultaneously decrease Ras localization to the PM and its lateral clustering, thus complicating data interpretation. As shown in Fig.S2I, the gold labeling intensities varied from a minimum of 44 gold particles to 1387 gold particles per $1\mu\text{m}^2$ PM area, a >30-fold difference. In our original manuscript, the gold labeling range was between 71 and 558 gold particles per $1\mu\text{m}^2$ PM area (from both untreated and treated cells). Thus, we now have a wider range of gold labeling than initially shown. Fig.S2I clearly shows that the L_{max} value is completely independent of gold labeling intensities, with an R^2 value of 0.001337.

Our data is also consistent with previous studies (Tian, et al. *Nat Cell Biol*, 2007 and Plowman, et al. *PNAS*, 2005) in showing no correlation between clustering and labeling densities. A recent study by Lee, et al. *bioRxiv*, 2019 uses single particle tracking in live cells and nicely illustrates that K-Ras lateral spatial distribution, including diffusion coefficients and clustered fraction, on the PM of live cells are completely independent of labeling densities. They altered the levels of the stably expressed photo-activatable mCherry (PAmCherry)-tagged K-Ras^{G12D} in the U2OS cells under doxycycline (dox) regulation. They compared the extent of PM localization of PAmCherry-K-Ras^{G12D} with the endogenous K-Ras. Their labeling densities varied from well below endogenous level of <10 molecules/ μm^2 of PM area, ~60 molecules/ μm^2 of endogenous labeling density, to high levels of ~300 molecules/ μm^2 (Lee, et al. *bioRxiv*, 2019). And they found that the dynamic and spatial behaviors of K-Ras are completely independent of labeling densities from well-below endogenous level to over-expressed levels (Lee, et al. *bioRxiv*, 2019). Our EM-spatial analysis also shows comparable and wider range of gold labeling (44-1387 gold labeling/ μm^2) and consistently show no correlation between clustering and labeling densities.

5. I'm not yet convinced about the biological relevance. What % of total area/how often are these curvatures observed on the plasma membrane of cells? In what context? If most Ras still locates to relatively flat areas of cells does this dominate anyway? I can imagine that these areas of curvature are more common on organellar surfaces and so may influence Ras trafficking/endomembrane localisation although this isn't discussed.

Our new AFM topography (Fig.S1) clearly illustrates that the cell surface possesses a wide range of curvatures. Thus, our observed curvature responses of Ras spatiotemporal organization suggest a highly heterogeneous behavior of Ras functions on the cell surface with differentially curved PM. The distinct membrane curvature preferences of different Ras isoforms may also contribute to their widely observed isoform-specific spatial segregation on the PMs.

We agree with the reviewer that the intracellular compartments possess highly varied curvatures. Specifically, Golgi apparatus contain highly curved intercisternal tubules (diameter ~20nm) and relatively flat disk-like cisternae. Endosomes are typically quite large, with diameters ~500nm. The curvature preferences of Ras isoforms may significantly influence Ras trafficking/endomembrane localization. It has been widely observed that Ras isoforms favor distinct endomembrane compartments. Specifically, H- and N-Ras localize to the Golgi, while K-Ras preferentially localizes to the recycling endosomes. Because these intracellular compartments possess distinct curvatures, the membrane curvature preferences of Ras isoforms may contribute to their distinct endomembrane localization and trafficking.

We have added new discussion on these points in the Discussion section (Pages 17-19).

Other comments:

1. Differences in membrane curvature also mean differences in membrane tension. Can the authors discriminate between these possibilities?

The reviewer has raised a very important issue. Bilayer curvature fluctuations and membrane tension have been intertwined significantly. Changing local curvature results in changes in the packing of lipid headgroups and acyl chains, and may in turn alter membrane tension.

However, we would like to note that lipid bilayers are much more resistant to stretching than bending. Because lipid bilayers are very thin, with a thickness of ~5nm, they can be easily bent. This is, indeed, reflected by the low bending stiffness parameters of synthetic lipid bilayers between 3×10^{-20} J and 9×10^{-20} J (Rawicz, et al. *Biophys J*, 2000; Needham and Nunn, *Biophys J*, 1990; Zhou and Raphael, *Biophys J*, 2005, Zhou and Raphael, *Biophys J*, 2007). Thus, lipid bilayers spontaneously undergo curvature fluctuations at room temperature, which can be modulated by osmotic stresses (Ho, et al. *Langmuir*, 2016). This is the basis for our osmotic stress experiments using GPMVs (Fig.3B and C). On the other hand, the lateral elasticity modulus, which characterizes the ability of a bilayer to stretch, is ~200mN/m. Thus, lipid bilayers are much more resistant to stretching.

Our own osmotic stress experiments using GPMVs also support this view. As described above, we used osmotic stresses to modulate the curvature fluctuations of the thin GPMV bilayers. While hypertonic stresses draw water out of the vesicles and induce more bilayer folding and wrinkles, hypotonic stresses pump water into vesicles and the resulting swelling unfolds and/or stretches the bilayer. Thermodynamic calculations have shown that hypertonic stress-induced membrane deformation is almost exclusively dominated by bending of the bilayer, without changing membrane tension (Ho, et al. *Langmuir*, 2016). Thus, hypertonic stresses correlate with the curvature regime. On the other hand, hypotonic stretching induces both bending and stretching, with the larger hypotonic stresses causing more stretching and more dominated by membrane tension.

Our Fig.3C shows that tK oligomerization responded to osmotic stresses in distinct manners. In the hypertonic and low hypotonic regimes (dominated by the changing bilayer curvature), tK oligomerization favors less folded and flatter membranes. However, at high hypotonic stresses ($\Delta\text{Osm} > 130\text{mM}$) when the membranes were extensively stretched and membrane tension was large, K-Ras anchor oligomerization was disrupted. Taking these data together, we can speculate that K-Ras oligomerization responds to membrane curvature and membrane tension in distinct manners: K-Ras clusters more extensively on less curved membranes but disassembles when the membrane tension is high. We need to perform additional

mechanical measurements to evaluate this. Our group is currently pursuing this, but we feel that this is out of the scope of the current study.

2. A topological diagram showing the different positive/negative orientations/radii of curvature induced by the different treatments would help the reader.

We have included a new table summarizing our data showing different orientations and magnitudes of curvature.

3. GFP-tagged Ras isoforms differentially locate to filopodia with more obvious KRAS localisation. This seems at odds with the interpretation here that HRAS favours areas of high curvature.

A possible explanation would be that Ras isoforms respond to different curvature directions in distinct manners. Filopodia protrude outward and away from the cell body. Ras proteins localized to the inner leaflet of the filopodia would be exposed to a highly convex surface with negative curvature.

We tested how Ras nanoclustering responded to membrane curvatures of different directions. We compared positive curvature-inducing BAR_{amph2} and BAR_{FCC} domains with the negative curvature-inducing inverse BAR_{IRS53p} domain in an EM-nanoclustering experiment. Inverse BAR domains, such as BAR_{IRS53p}, play important roles in filopodia formation (Frost, et al. *Cell*, 2009). As shown in our Fig.2H, K-Ras nanoclustering was disrupted by the positive curvature-inducing BAR domains, but was not affected by the negative curvature-inducing inverse BAR_{IRS53p} domain. On the other hand, H-Ras clustering was enhanced by the positive curvature, but was markedly disrupted by the inverse BAR_{IRS53p} domain. Thus, the negative curvature has no effect on the spatial distribution of K-Ras, but disrupts that of H-Ras. Based on our findings, we would expect that K-Ras localizes and clusters efficiently at PM with negative curvature while H-Ras spatial organization is markedly disrupted by the negative curvature within the filopodia. The distinct abilities of Ras isoforms to sense curvature direction may contribute to the higher K-Ras levels than H-Ras levels in filopodia.

4. The nanoclustering data in Supplementary Figure 1 appears to show that KRAS is nanoclustered over a wide size range suggesting that the labelling is so high that adjacent nanoclusters are also featuring in the analysis. The labelling seems too high if the influence of neighbouring nanoclusters is also featuring so strongly. The elevated condition (Supp Fig 1A/B) also reveals the potential for missing nanoclustering that may in fact be present. The labelling is so low that it is unlikely to report the presence of nanoclusters if they are there.

As described above, we performed additional EM experiments using different expression levels of GFP-K-Ras^{G12V} to vary the extent of gold labeling on the PM of untreated cells. Our new Fig.S2I shows no correlation between nanoclustering and gold labeling densities from ~44 gold particles / 1 μ m² PM area to ~1400 gold particles / 1 μ m² PM area.

Our data is also consistent with previous studies (Tian, et al. *Nat Cell Biol*, 2007 and Plowman, et al. *PNAS*, 2005) in showing no correlation between clustering and labeling densities. A recent study by Lee, et al. *bioRxiv*, 2019 uses live cell single particle tracking and nicely illustrates that K-Ras lateral spatial distribution, including diffusion coefficients and clustered fraction, on the plasma membrane of live cells are completely independent of labeling densities. They altered the levels of the stably expressed photo-activatable mCherry (PAmCherry)-tagged K-Ras^{G12D} in the U2OS cells under doxycycline (dox) regulation. They compared the extent of plasma membrane localization of PAmCherry-K-Ras^{G12D} with the endogenous K-Ras. Their labeling densities varied from well below endogenous level of <10 molecules/ μ m² of plasma

membrane area, ~ 60 molecules/ μm^2 of endogenous labeling density, to high levels of ~ 300 molecules/ μm^2 (Lee, et al. *bioRxiv*, 2019). And they found that the dynamic and spatial behaviors of K-Ras are completely independent of labeling densities from well-below endogenous level to over-expressed levels (Lee, et al. *bioRxiv*, 2019). Our EM-spatial analysis also shows comparable and wider range of gold labeling and consistently show no correlation between clustering and labeling densities.

Reviewer #3 (Comments to the Authors (Required)):

This paper provides a comprehensive report on studies of full length Ras isoforms and C-terminal lipidated peptides on different membrane settings; specifically it is found that K-Ras prefers a level/plane membrane whereas H- and N-Ras prefer a curved membrane. Membrane topology is influenced in several ways. This paper is of general interest for a wide community of protein-membrane scientists and of critical importance and timeliness for the Ras community.

The main points of the paper are strongly supported by the data throughout.

We thank the reviewer for the kind words.

This reviewer has several stylistic issues in that overall the paper is very dense and difficult to read. It should also be considerably shortened and possibly reorganized. A table summarizing all the results presented newly in the paper and those cited from the literature would greatly help.

We have included a new table summarizing all our data, as well as literature data. We also included two new heatmaps summarizing the clustering data.

Some of the language/terms are a little difficult for non-experts to get used to: esp. flatter cells/elongated shapes (where elongated actually means more curvature at the edges). Again, perhaps a table with all the membrane manipulations and outcomes would be useful. Similarly, an overview figures, in the spirit of a table of content schematic would be helpful to visualize the key finding.

We agree with the reviewer. In our manuscript, the flatter cells tend to have lower curvature, whereas cells with elongated shapes tend to have higher curvature. Indeed, micropatterning experiments found that flatter cells tend to possess more stimulated MAPK signaling, more growth and proliferation. On the other hand, elongated cells tend to possess less growth and proliferation. This is largely consistent with our findings. We have included a new table to summarize our data.

Recent literature which shows interactions between the K-Ras G-domain and PIP2 both computationally and experimentally should be referenced somewhere. The comment of a C-Raf membrane PS-binding domain is made but could be more explicit, saying what this domain is. Also, different computational labs. have reported different synergistic/non-synergistic binding effects between the Ras G-domain and the c-Raf CRD.

We thank the reviewer for reminding us the exciting findings of Cao, et al. 2019. We have added text to discuss the potential implications of our current work in the context of the exciting findings of Cao, et al. 2019. In particular, we speculate that the G-domain of K-Ras may contribute to membrane curvature sensing. Specifically, Cao, et al. 2019 illustrates that K-Ras G-domain more preferentially associates with PIP₂. Since we, here, show that PIP₂ distribution favors more curved membranes (opposite of PS), K-Ras G-domain may compete with its HVR (PS specific) for curvature sensing.

We have also included more text on the location of the PS-specific binding domain on C-Raf (Page 7). Specifically, the PS-binding motif on CRAF has been identified to be located in the cysteine-rich domain, amino acids 1-330 (Ghosh, et al. *J Biol Chem.* 1996), and more in particular amino acids 138-187, including hydrophobic residues L147, L149, L159 and L160 and charged residues R143, K144, K148 and R164 (Li, et al. *ACS Cent Sci.* 2018).

The synergistic/non-synergistic binding effects of the K-Ras G-domain and the CRAF CRD has also been discussed in our Discussion section (Pages 19-20). In particular, a recent work of Li, et al. 2018 predicts that, in a signaling complex of K-Ras with CRAF, CRAF CRD and K-Ras G-domain associate with lipid bilayers in a mutually exclusive manner. Thus, the combined K-Ras G-domain and CRAF complex may adopt a distinct orientation and present a membrane-interacting surface with a defined curvature, to which a lipid bilayer with a unique curvature may match.

The discussion section falls a little flat on contemplating the impact of the work. Specifically what are the biological rationales / implications of the different lipid / membrane shape preferences for Ras targeting to membranes and Ras extraction from the membrane or vesicle formation for recycling?

We have extensively modified our discussion section to speculate the potential impact of our work in the context of the isoform-specific Ras trafficking and intracellular distribution (Pages 18-20). In particular, Bastiaens and colleagues have proposed that all Ras isoforms spend considerable time as soluble and sample various endomembrane surfaces (Schmick, et al. *Cell* 2014; Schmick et al. *Trends Cell Biol.* 2015). As endomembrane compartments possess distinct and widely diverse curved surfaces, Ras membrane curvature sensing capabilities may contribute to the distinct preferential intracellular localization of different Ras isoforms (Golgi for H- and N-Ras, but recycling endosomes for K-Ras).

July 1, 2019

RE: Life Science Alliance Manuscript #LSA-2019-00343-TR

Prof. Yong Zhou
University of Texas Health Science Center at Houston
Integrative Biology and Pharmacology
6431 Fannin Street
MSE R382
Houston, TX 77030

Dear Dr. Zhou,

Thank you for submitting your revised manuscript entitled "Membrane curvature sensing of the lipid-anchored K-Ras small GTPases". As you will see, the reviewers appreciate the changes introduced in revision, and we would thus be happy to publish your paper in Life Science Alliance pending final revisions necessary to meet our formatting guidelines:

- please upload all figure files (main and suppl figures) as individual files, the figure legends should remain in the main manuscript docx file
- please link your profile in our submission system to your ORCID iD. You should have received an email with instructions on how to do so

A. FINAL FILES:

-- Summary blurb (enter in submission system): A short text summarizing in a single sentence the study (max. 200 characters including spaces). This text is used in conjunction with the titles of papers, hence should be informative and complementary to the title. It should describe the context

and significance of the findings for a general readership; it should be written in the present tense and refer to the work in the third person. Author names should not be mentioned.

B. MANUSCRIPT ORGANIZATION AND FORMATTING:

Sincerely,

Andrea Leibfried, PhD
Executive Editor
Life Science Alliance
Meyerohofstr. 1
69117 Heidelberg, Germany
t +49 6221 8891 502
e a.leibfried@life-science-alliance.org
www.life-science-alliance.org

Reviewer #1 (Comments to the Authors (Required)):

The authors have addressed satisfactorily each of my concerns and those of the other reviewers. The manuscript is ready for publication.

Reviewer #2 (Comments to the Authors (Required)):

I appreciate the careful and detailed responses by the author to my comments and many of the similar points also raised by reviewer 1. I am satisfied with the majority of the rebuttal that provided further experimental evidence to firm up conclusions related to eg. membrane tension, the role of the cytoskeleton and potential expression-related artefacts.

My core concern related to whether the observations could be linked to membrane curvature since there were no direct measurements of membrane curvature/the EM conditions may not have retained the induced curvature and associated nanoclustering. I am still not fully satisfied with the response; however, I also appreciate the difficult challenge of providing evidence directly correlating nanoclustering with regions of different curvature. The authors have now provided supplementary support to show that the conditions induce varying degrees of cell surface roughness and cite experiments where caveolae retain their curvature in the EM analysis. However, caveolae have a distinctive morphology that can be seen in the EM, whereas the EM sheets that were analyzed in this study have no visible evidence of curved profiles. Therefore, whilst the BAR domains are clearly inducing an effect on nanoclustering, it remains unclear if this is an indirect effect of BAR-domain expression or whether the majority of the gold particles analyzed are localized to BAR domain-induced curved areas that just look flat in the EM.

I am supportive of this manuscript and I recognize that there are multiple supporting lines of data that are consistent with the interpretation. I also know that it will be very difficult for the authors to provide the formal correlative proof of nanoclustering vs curvature status that I would ideally like to see, therefore the current submission is likely to be as refined as it can be. I'm sure that the manuscript will provoke broad interest and debate if published and will stimulate other groups to address these questions.

July 2, 2019

RE: Life Science Alliance Manuscript #LSA-2019-00343-TRR

Prof. Yong Zhou
University of Texas Health Science Center at Houston
Integrative Biology and Pharmacology
6431 Fannin Street
MSE R382
Houston, TX 77030

Dear Dr. Zhou,

Thank you for submitting your Research Article entitled "Membrane curvature sensing of the lipid-anchored K-Ras small GTPases". It is a pleasure to let you know that your manuscript is now accepted for publication in Life Science Alliance. Congratulations on this interesting work.

*****IMPORTANT:** If you will be unreachable at any time, please provide us with the email address of an alternate author. Failure to respond to routine queries may lead to unavoidable delays in publication.*******

DISTRIBUTION OF MATERIALS:

Again, congratulations on a very nice paper. I hope you found the review process to be constructive and are pleased with how the manuscript was handled editorially. We look forward to future exciting submissions from your lab.

Sincerely,
